# Increasing wintertime cloud opacity increases surface longwave radiation at a long-term Arctic observatory

Leah Bertrand [1,2] ✉, Jennifer E. Kay [1,2,4] ✉ & Gijs de Boer[3,4]

As the Arctic warms, winter clouds are known and expected to change. Yet the extent to which these cloud changes amplify or dampen warming (cloud feedback) remains uncertain. This uncertainty results from systemic difficulties in modeling and observing Arctic low clouds. Surface-based observations avoid many of these difficulties. Here, we use two decades of surface-based observations (1998–2023) to constrain and explain longwave flux change during winter. We find that longwave flux into the surface is increasing and that this increase cannot be explained by direct impacts of temperature and greenhouse gases alone. Only when increasing cloud radiative effect $(0.96 \pm 0.64 \text{ W/m}^2/\text{K})$ is considered can increasing longwave flux be explained. Cloud radiative effect increases due to increasing cloud opacity, which is driven equally by ice-only and mixed-phase clouds. The direct observational constraint from this work suggests that increasing cloud opacity drives increasing net surface radiation on Alaska's North Slope during winter.

The Arctic surface and lower atmosphere have warmed dramatically in recent decades, especially during fall and winter[1]. Clouds are known to respond to this warming[2–5], but quantification of cloud impact on climate remains challenging[6–9]. Cloud feedback refers to how changes to clouds redistribute energy in the climate system, thereby amplifying or dampening surface warming. Arctic cloud feedbacks cannot be confidently assessed from models, due in large part to unconstrained cloud microphysical parameterizations[6,10–16]. Observational estimates of cloud feedbacks avoid this structural uncertainty, offering an attractive alternative approach.

Observational estimates of cloud feedbacks can be based on satellite measurements from top-of-atmosphere (TOA)[17–19], but TOA radiation alone is insufficient for the Arctic. Low clouds can have opposite effects on TOA and surface radiation[20,21], making for potentially competing feedback effects. This relative independence of TOA and surface radiation (Supplementary Fig. 1) means that satellites struggle to retrieve cloudy-sky Arctic surface radiation[22–24], which complicates assessment of Arctic cloud feedbacks. This fundamental difficulty has improved with active remote sensing satellites that

observe cloud vertical structure, but even they still underestimate Arctic low clouds and their effect on surface radiation[25–28].

Due to the challenges of using satellite observations or models alone, there is a pressing need to develop ground-based observational constraints on cloud-radiative feedbacks in the Arctic. Ground-based observations are frequently used to validate models and satellite retrievals[26,29–32], but they are rarely used to study cloud feedbacks directly because of their typically short-term records. Yet an Arctic observatory on the North Slope of Alaska (NSA) has taken decades of extensive observations in the context of local warming. Operating since 1998, the Atmospheric Radiation Measurement (ARM) program's NSA facility[33] offers a unique opportunity to directly investigate Arctic cloud feedbacks using ground-based measurements. We use 26 years (1998–2023) of surface radiation measurements and 13 years (2004–2019) of comprehensive cloud measurements. This record length is comparable to or longer than existing cloud feedback studies[17,19,34].

Even though ground-based observations from this single site cannot directly sample the entire Arctic, they sample air masses

[1]Department of Atmospheric and Oceanic Sciences, University of Colorado, Boulder, Boulder, CO, USA. [2]Cooperative Institute for Research in Environmental Sciences, University of Colorado, Boulder, Boulder, CO, USA. [3]Environmental and Climate Sciences Department, Brookhaven National Laboratory, Upton, NY, USA. [4]These authors jointly supervised this work: Jennifer E. Kay, Gijs de Boer. ✉e-mail: leah.bertrand@colorado.edu; jennifer.e.kay@colorado.edu

advected from a wider area and provide a powerful constraint at a single location. For ARM NSA, air masses are advected from both sea ice and land due to prevailing northerly winds[35]. At only 2 km away from the coast, ARM NSA lies at the intersection of sea ice and continental climate regimes. These two regimes have weakened contrast during winter, since both are frozen and snow-covered.

Our analysis is limited to winter alone (December–March) because it has the greatest surface warming without surface melt (99.9% of near-surface air temperature observations <−0.5 °C, Supplementary Fig. 2). Locally snow-covered land and nearby pack ice respond similarly to atmospheric forcing during winter[36,37], suggesting that a wintertime constraint developed at ARM NSA would be representative of a larger area. These two surface types, since they are both frozen, have freely changing surface skin temperatures in response to atmospheric (e.g., cloud) radiative forcing. Furthermore, wintertime non-radiative fluxes only weakly influence the overall response to atmospheric forcing[37]. This simplifies the surface energy budget to be dominated by net radiative flux, which is limited to longwave (i.e., thermal emitted) radiation only due to polar night. Surface-atmosphere interactions are then tracked by net surface longwave radiation (hereafter net surface flux, downward positive), which provides a window into cloud-induced warming.

Net surface radiative flux is important because it cools the Arctic via loss of energy to space when negative[38–40]. It is the key quantity distinguishing the two radiative states of the wintertime Arctic[41]. Yet net surface flux response to warming is unknown – it can either increase or decrease with warming due to competing physical processes. Warming alone implies a decrease in net surface flux, as more energy is emitted to space via the Planck feedback. Many other responses to warming can increase net surface flux, such as increasing water vapor, cloud cover, or cloud opacity. Owing to the unique characteristics of Arctic winter, there are no a priori expectations for the wintertime cloud response to warming.

In this work, we leverage ARM NSA's rich and lengthy measurement record to observe and disentangle competing cloud and non-cloud responses to warming. Using local atmospheric warming and variability, we estimate the local surface radiative feedback from observations and diagnose its cloud and atmospheric drivers. We find that cloud changes drive the overall radiative response to warming. Increasing cloud opacity determines that less–rather than more–energy leaves the surface with warming.

## Results
### Observed changes in surface radiative fluxes
Like the Arctic as a whole, the environment around the ARM NSA facility is warming rapidly. Specifically, from 1998 to 2023 NSA observations show surface warming of 0.9 ± 0.5 K/decade (95% confidence). Observed longwave radiation is consistent with this warming, as both upward and downward radiative fluxes are increasing over time (Fig. 1a). Yet the difference of these two terms–the net surface flux (downward minus upward)–has no significant time trend. No trend is detected because the impacts of surface and atmospheric warming have opposite effects on the net surface flux, making the net change smaller than the upward or downward change alone.

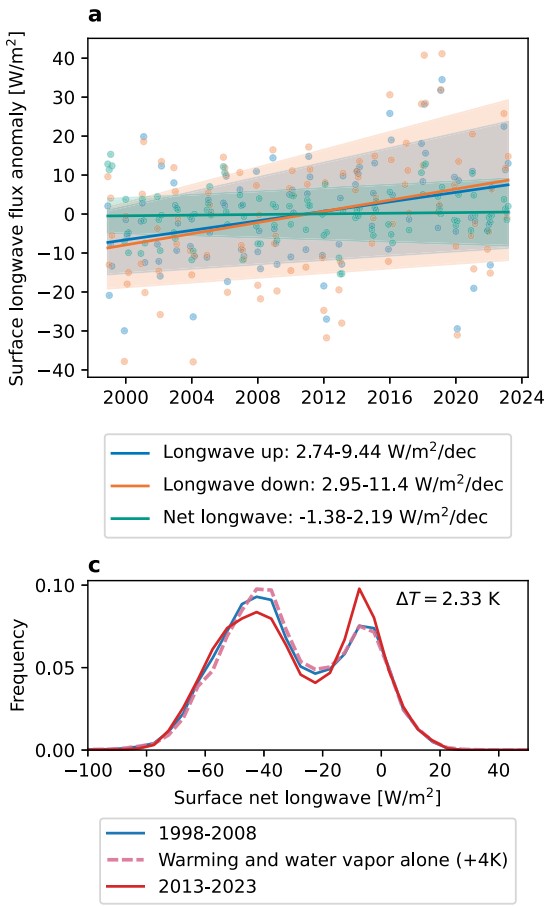
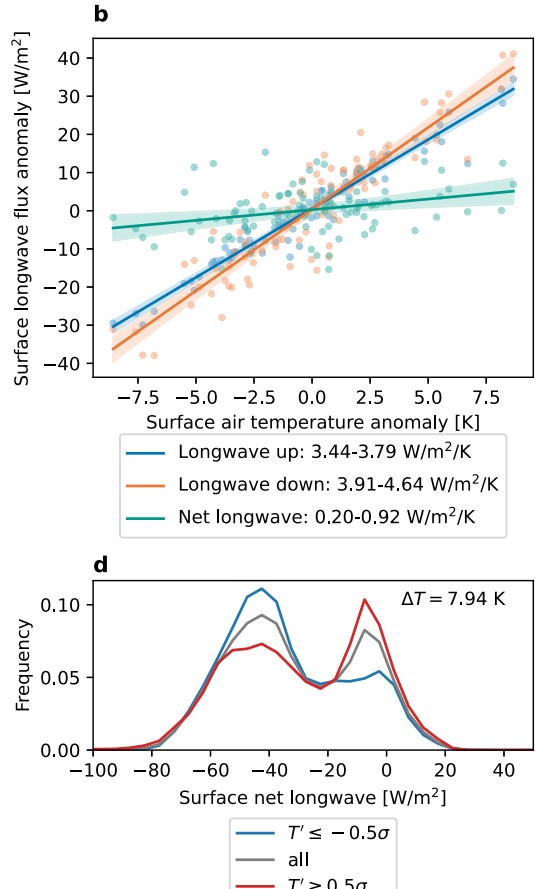

**Fig. 1 | Observed changes in wintertime surface longwave fluxes. a** 26-year trends in wintertime surface longwave fluxes and **b** surface longwave flux response to monthly surface air temperature anomalies. Regression 95% confidence band shown in shaded area and reported in legend. **c** Net surface longwave flux (downward positive) distribution change over time and expected change from 4 K warming at constant relative humidity. **d** Net surface longwave flux distribution composited by monthly surface air temperature anomaly $T'$ ($\sigma = 3.4$ K). Text ($\Delta T$) in (**c**) and (**d**) indicates the average temperature difference between composites.

By regressing against time, large variability in net surface flux is noise obscuring the signal (i.e., response to warming). But by instead regressing against monthly temperature anomalies, the signal is clearer. The flux response to temperature variations combines warming and variability, giving the short-term radiative feedback[17]. This approximation of the radiative response to warming alone revealed net surface flux increased with temperature from 1998 to 2023 (0.20–0.92 W/m²/K, 95% confidence). Net surface flux increased because the downward flux had a larger increase than the upward flux.

In addition to the monthly mean, the net surface flux distribution also provides valuable insight. The bimodal shape of the net surface flux distribution (Fig. 1d, gray) is a well-known and important feature of the wintertime Arctic[41]. It shows the bifurcation of the Arctic into radiatively clear (−50 W/m²) and opaquely cloudy (0 W/m²) radiative states. The radiatively clear state is associated with cold air and thin ice clouds or clear skies, while the opaquely cloudy state is associated with warmer air and overcast mixed-phase clouds[15,41–45]. Some studies have used indirect evidence to suggest that opaque mode frequency may increase with warming in the Atlantic sector of the Arctic[44,46], but they did not directly observe changes to net surface flux. While well-characterized in the climatological sense, the response of this distribution to warming is unknown.

A key result is that the net surface flux distribution shows a shift from the clear to the opaque mode with warming (Fig. 1c, d). This pronounced shift to the opaque mode may be driven by clouds. Clouds are implicated in this change because warming and increasing greenhouse gases alone (Section "Radiation distribution shape change attribution") have no effect on opaque mode frequency (Fig. 1c, dashed line). Warming has no direct impact because the opaque mode is already in surface-atmosphere radiative equilibrium. Increasing greenhouse gases have no direct impact because of masking by low clouds. Increasing opaque mode frequency suggests changing cloud properties may explain surface longwave flux change.

Opaque mode frequency increases whether distributions are composited by time (Fig. 1c) or monthly-mean temperature anomaly (Fig. 1d). Because of local warming, the most recent 10 winters of observations (2013–2023) are 2 K warmer than the first 10 winters (1998–2008) (Fig. 1c). Because of local temperature variability, the highest 31% ($\geq 0.5\sigma$) of monthly temperature anomalies is 8 K warmer than the lowest 31% ($\leq -0.5\sigma$) (Fig. 1d). As before, the opaque mode increase is more pronounced using temperature anomaly as a coordinate. The relative position of the two modes in the distribution is the same in all composites, suggesting its persistence in a changing climate.

## Drivers of increasing downward radiation

Increasing downward radiation drove the observed increase in net surface radiative flux at ARM NSA (Fig. 1). But why did downward radiation increase more than upward radiation? Upward radiative flux response to warming is dominated by the Planck feedback, with the observed increase explained entirely by this simple expectation (Supplementary Note 1, Supplementary Fig. 3). This simplicity implies that other surface fluxes (sensible and latent heat flux, subsurface heat flux) are at most secondary drivers and not discussed further. To understand increasing net surface flux, we must investigate the increase in surface downward longwave flux.

Downward longwave flux has a complex dependence on a wide range of cloud and atmospheric properties. Fortunately, ARM NSA has all the long-term observations needed to explain increasing downward flux. Using radiosondes and a multi-sensor cloud microphysics retrieval[47], we examine how radiatively important properties (drivers) change with warming. Supplying these observations to a radiative transfer model, we isolate the radiative effect of each changing driver individually (Section "Radiation driver attribution"). Increasing downward flux with warming is then attributed to changes in 4 non-cloud (Planck, Water vapor, Direct $CO_2$, and Lapse rate in Fig. 2e) and 4 cloud (Cloud water path, Cloud phase, Cloud cover, Cloud altitude in Fig. 2e) drivers. Each driver is furthermore divided according to observed cloud phase type - clear skies, mixed-phase clouds, liquid-only clouds, or ice-only clouds (Section "Decomposition by cloud phase").

Within this set of 8 drivers, we have clear expectations for the impacts of increasing temperatures and greenhouse gases. We expect Planck and water vapor to drive the largest non-cloud increases in surface downward longwave radiation. Cloud drivers, on the other hand, have no such a priori expectations. Increased moisture availability from warming increases cloud water path[48–51], but the rate of this increase depends on poorly-constrained microphysical processes. Warming generally increases the frequency of cloud liquid relative to ice[52], but phase transition is only weakly constrained when temperatures do not cross 0 °C (ref. 53 Fig. 10). In short, many possible driver responses could occur and the relative importance of each is unclear.

With our attribution methodology, we can directly observe all of these drivers and rank them by importance of their radiative impact. Temperature and water vapor are strongly positive drivers. The sum of all non-cloud drivers explains 76% of the total attributed flux change (3.08 ± 0.56 out of 4.04 ± 2.06 W/m²/K). The remaining 24% (0.96 ± 0.64 W/m²/K) is due to many different changing cloud properties. Yet increasing cloud opacity – via increasing cloud water path and ice-to-liquid cloud phase change – drives the largest cloud increase in surface downward longwave radiation.

Cloud opacity-induced radiation increases are driven equally by ice-only and liquid-containing clouds. Ice-only clouds drive a 0.44 ± 0.06 W/m²/K increase via the conversion of thin to opaque ice clouds (i.e., ice-only component of cloud water path driver in Fig. 2e). Liquid-containing clouds drive a 0.43 ± 0.60 W/m² increase through both conversion of thin to opaque liquid-containing clouds (i.e., mixed-phase and liquid-only components of cloud water path driver) and ice-to-liquid phase change (i.e., cloud phase driver). The large role played by ice-only clouds suggests that ice-only cloud processes have similar importance to mixed-phase cloud processes in determining the cloud feedback during the cold winter at NSA.

Limiting our focus to the thin-to-opaque cloud transition alone, we find that ice-only clouds drive most (77%) of the overall cloud water path effect. The thin-to-opaque transition has a larger impact in ice-only clouds because ice clouds are less opaque than liquid-containing clouds on average (Fig. 2b–d). In other words, ice-only clouds have a larger margin via which to increase radiation without a change in cloud phase. The lower mean-state opacity of ice clouds results in a larger gray- to blackbody transition that has substantial longwave consequences.

## Clouds explain increasing net surface flux

Now that changes in upward and downward longwave radiation have been explained, we can investigate the drivers of changing net surface longwave flux. Increasing cloud opacity is the key driver of increasing net surface flux, since the observed radiation change cannot be explained without it (Fig. 3a). If only warming occurred while everything else remained constant (Planck in Fig. 3), net surface flux would decrease instead of increase. Further, if only cloud properties remained constant while everything else was allowed to change (+Non-cloud), net surface flux would still decrease. Only when changing cloud properties are added (+Cloud) can the independently observed increase (Total) be explained. In other words, non-cloud feedbacks alone are insufficient to explain observed increasing net surface flux.

Separating net surface flux change by cloud phase type (Fig. 3c–e) highlights the importance of increasing cloud water path. Within a single cloud phase, cloud cover and phase changes are not considered. Only cloud water path and cloud altitude remain as cloud drivers in these subsets. Still, the impact of increasing cloud

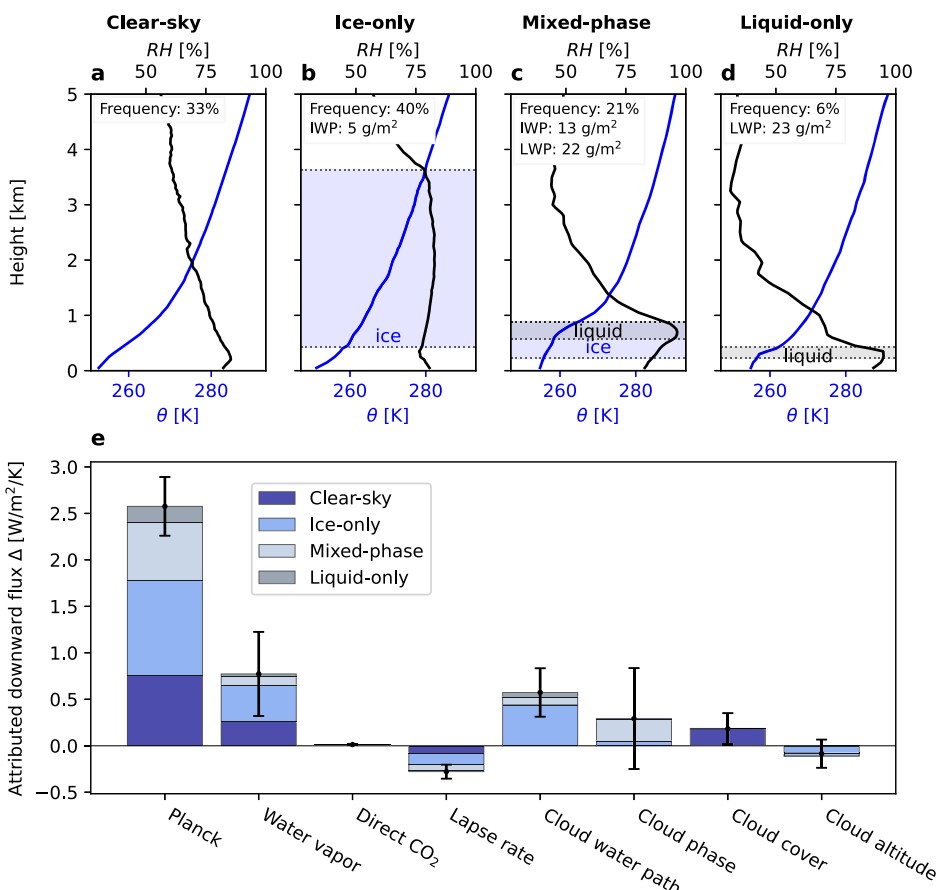

**Fig. 2 | Quantifying drivers of increasing downward longwave flux with warming. a–d** Median lowest cloud layer boundaries (dotted lines and shaded area), potential temperature ($\theta$), and relative humidity with respect to liquid water (*RH*) by cloud phase type. Text in panels indicate frequency of each cloud phase type and liquid and ice water path (LWP and IWP) as applicable. **e** Surface downward longwave flux change ($\Delta$) attributed to observed changes in each driver individually. Drivers are 4 non-cloud properties (Planck, Water vapor, Direct $CO_2$, Lapse rate) and 4 cloud properties (Cloud water path, Cloud phase, Cloud cover, Cloud altitude) which explain the radiation change. All-sky attribution is a weighted sum of driving by each cloud phase type. Bars indicate all-sky 95% confidence intervals (calculated as described in Section "Uncertainty analysis").

opacity is pronounced. For each cloud phase type individually, the observed net flux change cannot be confidently explained without increasing cloud water path (+Non-cloud vs. Total in Fig. 3c–e). Further, net surface flux increases only during ice-only clouds because of their stronger change from thin to opaque clouds. The marked change under ice-only clouds highlights the importance of ice cloud processes for the overall surface radiative feedback during the cold winter at NSA.

Our driver attribution methodology relies only on cloud and atmospheric observations (Section "Radiation driver attribution"). In other words, drivers are attributed independently from directly observed radiation. Yet the attributed total radiation change (+Cloud) is consistent with directly observed totals (Total) for all conditions. This agreement means that our method achieves radiative closure. In other words, our method correctly predicts the total radiation change from independently observed changes in cloud and atmospheric properties for all conditions.

### Changes independent of circulation variability

Synoptic disturbances can drive transitions between clear and opaquely cloudy radiative states[42]. As such, short-term variations in the frequency of synoptic disturbances might confound the long-term cloud and radiation response to warming estimated above. For example, anomalously warm months may have more synoptic disturbances that would result in more opaque clouds. To control for this potential confounding, we leverage the established relationship between daily local surface pressure and radiative state in the wintertime western Arctic[35,41–43] (Supplementary Fig. 5). By repeating key regressions stratifying by daily local surface pressure, circulation-induced transitions between clear and cloudy states are largely removed. We find that both net radiation and cloud water path increases are not due to local surface pressure variability.

At all but extreme low pressures and weak high pressures, net surface flux increases with temperature (Fig. 4a). The magnitude of the increase is similar to the estimate without controlling for local surface pressure (Fig. 4b). In other words, net surface flux response to warming is generally independent of the circulation. As an additional line of evidence, there is a long-term warming trend but no local surface pressure trend 1998–2023 (Supplementary Fig. 4).

In addition to net flux response, its key driver—cloud water path—increases independently from circulation variability (Fig. 4c). For all but extreme high pressures, cloud water path increases are similar to or greater than the initial unstratified estimate (Fig. 4d). Controlling for a different circulation confounder (reanalysis-derived temperature advection) yields the same result of robustly increasing net flux and cloud water path (Supplementary Fig. 6). Since controlling for synoptic disturbances by two independent methods did not remove or reverse our initial estimate of either cloud water path change or net flux change, we conclude that circulation variability does not confound our analysis. In other words, variations in atmospheric circulation cannot explain the relationship between temperature, cloud opacity, and net surface flux reported here.

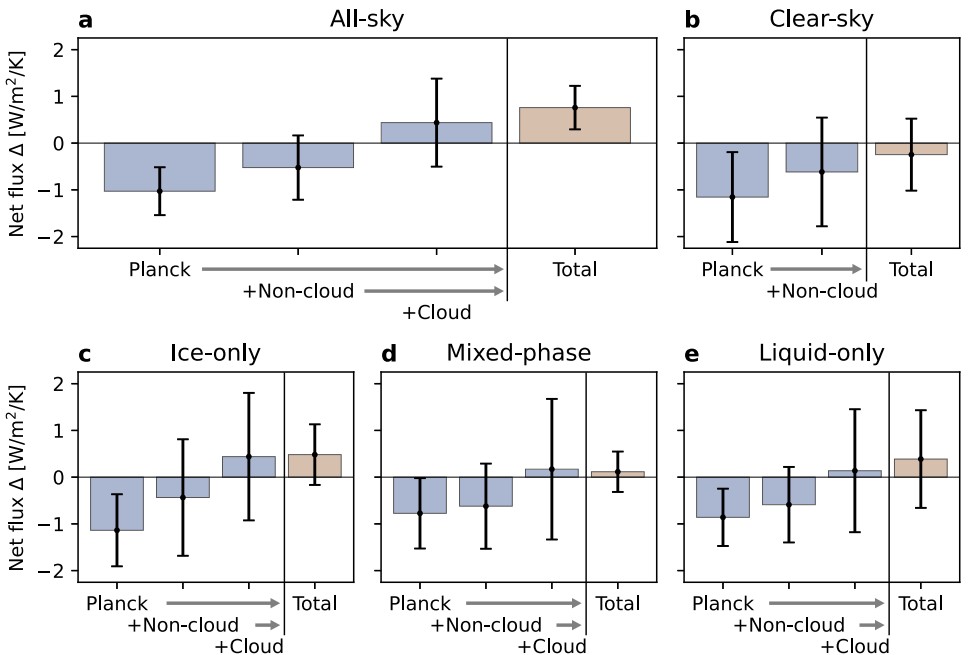

**Fig. 3 | Increasing net surface longwave flux is only possible with changing cloud properties.** Net surface flux response to warming (Δ) from cumulatively adding drivers for **a** All-sky, **b** clear-sky, **c** ice-only clouds, **d** mixed-phase clouds, and **e** liquid-only clouds. All-sky net flux change (**a**) includes changes in both cloud phase type frequencies and properties (Eq. (4)). Net flux change for an individual cloud phase type (**b**–**e**) only includes changing properties. Bars show expected net surface flux change for warming alone (Planck); warming, lapse rate, and greenhouse gas changes (+Non-cloud), and changes in all cloud and atmospheric properties (+Cloud). The total attributed response from cloud and atmospheric observations (+Cloud) should be compared to the total net surface flux response (Total) directly observed from radiometers. Bars indicate 95% confidence intervals.

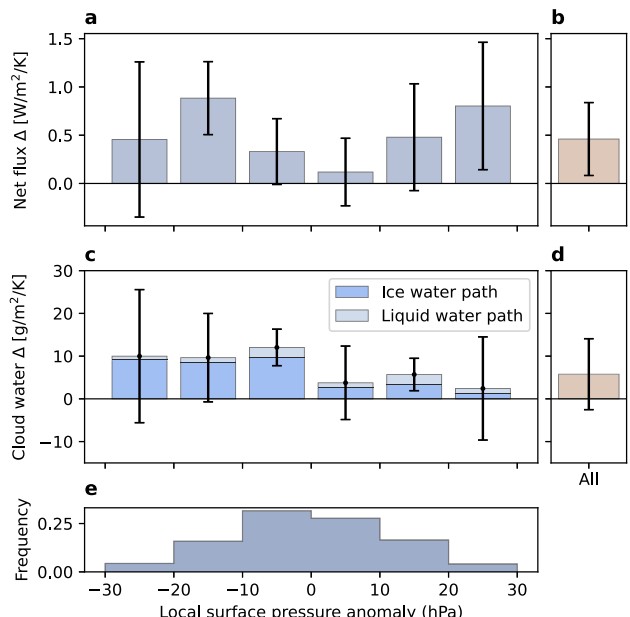

**Fig. 4 | Circulation variability cannot explain increasing net surface flux and cloud water path.** **a**, **b** Net surface flux and **c**, **d** cloud water path response to warming (Δ) with (**a**, **c**) and without (**b**, **d**) controlling for circulation variability. Circulation variability is removed by stratifying (binning) by daily local surface pressure anomaly. **e** Frequency of each local surface pressure anomaly bin. Bars show 95% confidence interval for regression slopes. **c** Total and individual ice and liquid contributions to cloud water path.

## Discussion

The main takeaway from this work is that, during winter, the surface of the North Slope of Alaska retains more energy with warming due to increasing cloud opacity. The local cloud feedback of $0.96 \pm 0.64$ W/m²/K cancels out the increased release of energy implied by the sum of all other (non-cloud) feedbacks. The total local radiative feedback at the surface is then positive. In other words, increasing surface temperatures do not increase surface radiative cooling because of changing cloud properties. This fact provides direct observational evidence that clouds contribute to surface-amplified warming during winter at NSA.

Increasing cloud water path was the largest driver of the cloud feedback ($0.57 \pm 0.26$ W/m²/K). Cloud water path increased in all cloud phase types, but ice-only cloud radiative effect increased more strongly because of its lower mean-state opacity. Increasing opacity of ice-only clouds drove the majority (77%) of the radiation increase from changing cloud water path. Dependence of the feedback on mean-state opacity suggests that constraining present-day cloud water path in thin ice clouds may be key to constraining the wintertime Arctic cloud feedback.

While an ice-to-liquid cloud transition occurred, it was not the largest cloud feedback driver. Cloud-phase change increased surface radiation by $0.29 \pm 0.54$ W/m²/K, while the thin-to-opaque ice cloud transition increased surface radiation by $0.44 \pm 0.06$ W/m²/K. The secondary role of cloud-phase change at NSA highlights the importance of ice processes during the cold winter. Even though conversion of ice to liquid has a clear physical explanation and has received much attention, it may have a smaller effect than increasing ice production during winter. Future work might develop along two lines of inquiry – what cloud processes drive increased ice production and increasing cloud opacity generally? Does a thin-to-opaque cloud transition occur in other seasons and regions of the Arctic, or is it unique to NSA winter?

## Methods

### Data

The ARM NSA facility is one of the most heavily instrumented ground-based observatories in the world[33]. Among the many measurements available, two datasets form the core of this study: surface longwave radiation (QCRAD[54]) and comprehensive cloud properties (Shupe-Turner microphysics[55]).

The first portion of this study (Section "Observed changes in surface radiative fluxes") relies only on QCRAD and surface meteorology[56,57]. The lengthy (1998–2023) and well-calibrated record captures how the radiation is responding to warming. In order to explain the response, however, more observations are needed.

In the second portion of this study (Section "Drivers of increasing downward radiation"), cloud and atmospheric observations are introduced to explain the changing radiation. Cloud observations come from a multi-sensor retrieval suite developed specifically for use in the Arctic –Shupe-Turner microphysics[55]. With 13 years of observations combining many instruments (radar, lidar, ceilometer, microwave radiometer, radiosonde), it offers the detail typically associated with a field campaign at the duration more typical of a satellite mission. Combined with temperature and humidity from radiosondes[58], it offers everything needed to explain the radiation.

### Radiative transfer

Cloud and atmospheric observations can be supplied to a radiative transfer model to enable a powerful tool to explain changes in the actual observed radiation. We use the Rapid Radiative Transfer Model–Longwave (RRTM-LW)[59] to calculate radiation from observed cloud and atmospheric state, as described in[55]. Calculations are done by supplying Shupe-Turner cloud microphysics and temperature/humidity profiles[58,60,61] to RRTM-LW. Since cloud and atmospheric state are observed independently from surface radiation, comparison provides a natural opportunity for validation.

By comparing calculated to observed radiation, we can check our ability to explain the radiation. Such a check is important because several instrument upgrades (radar and lidar) may affect the long-term explanatory power of observations. We compare a 20-minute rolling average of calculated radiation to observed radiation during the minute in the center of the averaging window[55].

Validation quality is stable over time, with consistent performance in every year (Fig. 5a). The median difference is near or less than that expected from radiometric observational uncertainty for every year (±5 W/m² for minute-frequency observations). The all-time median difference is −2.6 W/m², while the smallest (largest) is 0.1 (−6.1) W/m² in 2018 (2005). The interquartile range of differences is also generally near this expected ±5 W/m² spread. The all-time average interquartile range is 13.1 W/m², while the minimum (maximum) is 9.2 (18.8) W/m² in 2014 (2006). For each year (Supplementary Table 1), our degree of

closure is similar to existing state-of-the-art Arctic closure exercises (Supplementary Table 2)[24,55,62–64]. Consistent performance over time shows that the data are suitable for long-term analysis.

In addition to long-term stability, instantaneous variations in radiative flux are well-explained by cloud and atmospheric observations. Cloud and atmospheric observations explain 87% of the minute-frequency variance in observed radiation (Fig. 5b). Calculated and observed fluxes are evenly distributed along the 1:1 line from the coldest to warmest conditions. Good performance over all conditions suggests that cloud observations (and calculations) are suitable for explaining long-term changes in the surface downward longwave flux.

### Radiation distribution shape change attribution

Surface radiation responds to warming, as do cloud and atmospheric properties. To link these two, we perturb observed cloud and atmospheric properties and recalculate radiative flux in RRTM-LW. First, we assess whether increasing temperatures and water vapor alone can explain the change over time in the net surface flux distribution.

Surface longwave fluxes are recalculated with 4 degrees Kelvin added to surface and atmospheric temperatures for the entire data record. Water vapor increases with constant relative humidity. We find the change in the distribution of calculated net flux according to

$$\Delta n_k^{calc} = n_k(F_{calc}(\ +4K)) - n_k(F_{calc}(2004 - 2019)), \qquad (1)$$

where $n_k(F_{calc}(X))$ indicates the relative frequency of bin $k$ in the histogram of calculated net surface flux $F_{calc}$ under a condition $X$. Expected change in net flux distribution $\Delta n_k^{calc}$ is then added to the directly observed (not calculated) net surface flux distribution $n_k(F_{obs})$. Then $\Delta n_k^{calc}$ is added to the net flux distribution from the first decade of observations according to

$$n_k(F_{obs}(\ +4K)) = \Delta n_k^{calc} + n_k(F_{obs}(1998 - 2009)), \qquad (2)$$

where $n_k(F_{calc}(X))$ refers to the observed net surface flux histogram for a condition $X$. This perturbed distribution $n_k(F_{obs}(+4K))$ is then plotted in Fig. 1c and labeled as warming and water vapor alone (+4K). By comparing the expected change in the 1998–2008 distribution to the actual 2013–2023 distribution, it can be assessed whether warming and water vapor alone could be responsible for the net surface flux change over time.

### Radiation driver attribution

To further understand why surface longwave fluxes are changing, we find the radiation change due to changes in each cloud and atmospheric property individually. This approach allows us, for example, to isolate the effects of changing cloud properties on the radiation. We call these radiatively important cloud and atmospheric properties

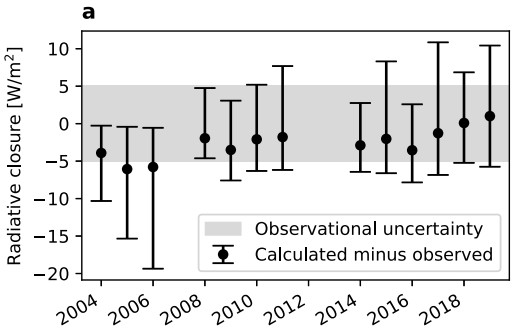
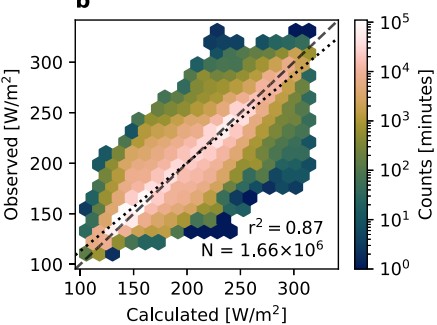

**Fig. 5 | Evaluation of cloud and atmospheric measurements.** Agreement between directly observed and calculated fluxes from cloud and atmospheric observations **a** over time and **b** over the entire data record. Bars in (**a**) show median and interquartile range of differences for DJFM. Shaded area indicates instantaneous measurement uncertainty of radiometers of ±5 W/m². Regression of calculated against observed radiation (dotted line) (**b**) and one-to-one line (dashed line).

drivers. We quantify 8 drivers – 4 non-cloud (Planck, lapse-rate, $CO_2$, water vapor) and 4 cloud drivers (cloud water path, cloud phase, cloud cover, cloud base height) – which explain the total surface downward longwave radiation change.

Driver effects are attributed through a first-order Taylor series approximation. For each driver $x_i$, its contribution to flux change with warming ($dF/dT$) is

$$\frac{dF}{dT}\Big|_{x_i} \approx \frac{\partial F}{\partial x_i}\frac{\partial x_i}{\partial T}. \tag{3}$$

The first term ($\partial F/\partial x_i$) is the radiative effect of a change in driver $x_i$, and the second term ($\partial x_i/\partial T$) is the change in driver $x_i$ with warming. The first term $\partial F/\partial x_i$ is found by perturbing a single driver $x_i$ in observed cloud and atmospheric profiles and calculating the radiative effect of the perturbation (i.e., sensitivity) with RRTM-LW. Profiles are taken from radiosonde launches and microphysics averaged in a 31-minute window centered on sonde launch time. The second term $\partial x_i/\partial T$ is found via ordinary least squares regression of monthly-mean anomalies in $x_i$ against monthly-mean anomalies in near-surface air temperature $T$.

### Decomposition by cloud phase

The radiative effect of changes in drivers depends on mean cloud and atmospheric state. For example, opaque clouds would be insensitive to liquid water path changes while thin clouds would be highly sensitive. Since cloud phase is a primary control on opacity, we calculate (3) separately for each cloud phase type. Driver attribution is then split into four states with distinct radiative responses to perturbations - clear skies, ice-only clouds, mixed-phase clouds, and liquid-only clouds.

We classify each observation into one of these four categories by aggregating the vertically-resolved hydrometeor phase classification in the Shupe-Turner product. Classification is simple for ice-only, since it is defined to be no liquid anywhere in the profile. Classification is less straightforward for liquid-only, since high ice clouds often overlie liquid-containing layers. A profile is classified as liquid-only if a cloud contains a liquid-only volume without ice at the boundaries of the liquid-containing cloud layer. If liquid and ice are present in the same cloud, it is classified to mixed-phase whether or not the liquid and ice co-occur at the same vertical level. These three cloud phase type definitions are mutually exclusive, so each cloudy profile is classified into only one category.

With observations separated into these four categories, we can decompose (3) by cloud phase type. The all-sky effect of changes in $x_i$ is a weighted mean of its effects in each state $j$:

$$\frac{dF}{dT} = \frac{dF}{dT}\Big|_{x_i} + \frac{dF}{dT}\Big|_{f_j} \approx \sum_{i,j} f_j \frac{\partial F_j}{\partial x_{ij}}\frac{\partial x_{ij}}{\partial T} + \sum_j \frac{\partial F}{\partial f_j}\frac{\partial f_j}{\partial T}, \tag{4}$$

where $f_j$ is the overall frequency of state $j$. Changes in the frequency of each state $j$ are added as drivers. Driving by changes in state frequencies $f_j$ ($dF/dT|_{f_j}$) is a product of two terms. The first term $\partial F/\partial f_j$ is calculated by resampling (bootstrapping) the observed distribution of radiation to have a more-frequent state $j$. This resampling gives the effect of a change in $f_j$ on all-sky average $F$. The second term $\partial f_j/\partial T$ is found by regressing monthly-mean anomalies in $f_j$ against monthly-mean anomalies in $T$.

Most drivers can be calculated from (4). Drivers evaluated from (4) are vertically constant warming (Planck driver), vertically constant warming minus observed vertical structure of warming (lapse rate driver), observed vertical structure of atmospheric moistening (water vapor driver), vertically constant $CO_2$ mixing ratio increase (direct $CO_2$ driver), cloud altitude decrease with constant cloud base potential temperature (cloud base height driver), and change overall cloud

cover without a change in cloud phase (cloud cover). Increasing cloud cover is diagnosed from the effect of decreasing $f_{clr}$.

### Uncertainty analysis

We report 95% confidence intervals for attributed $dF/dT$. These are calculated by propagating uncertainty from partial derivatives in Eq. (4). The two categories of terms are driver response to warming ($\partial x_i/\partial T$ and $\partial f_j/\partial T$) and flux sensitivity to driver ($\partial F/\partial x_i$ and $\partial F/\partial f_j$). For uncertainty in the driver response to warming, we use 95% confidence intervals on regression slopes evaluated using Student's $t$-distribution (note that all regression uncertainty in this study is evaluated this manner). For uncertainty in the flux sensitivity to driver, we use the 5th and 95th percentiles from the population of observed sensitivities. To convert 95% confidence intervals to scalar uncertainties, we assume $\delta = \frac{F_{cdf}(0.95) - F_{cdf}(0.05)}{2}$, where $F_{cdf}$ is a cumulative distribution function of an uncertain term and $\delta$ is the uncertainty in that term. Note that uncertainty in $dF/dT$ for Planck and Lapse rate drivers comes only from $\partial F/\partial x_i$ because temperature is the driver $x_i$ ($\partial x_i/\partial T = 1$).

### Cloud phase change attribution

To separate cloud cover changes from cloud phase changes, we define the relative frequency of cloud phase types $r_j$ as

$$r_j = \frac{f_j}{1 - f_{clr}}, \quad j \neq clr. \tag{5}$$

The cloud phase driver is the radiative effect of changes in the frequency of each cloud phase type relative to all clouds. This effect is calculated from

$$\frac{dF}{dT}\Big|_{phase} = (1 - f_{clr}) \sum_{j \neq clr} \frac{\partial F_{cld}}{\partial r_j}\frac{\partial r_j}{\partial T}. \tag{6}$$

We calculate $\partial F_{cld}/\partial r_j$ by bootstrapping cloudy-sky radiation to have a more-frequent state $r_j$. We calculate $\partial r_j/\partial T$ by regressing monthly-mean anomalies in $r_j$ against monthly-mean anomalies in $T$. Uncertainty is calculated for (6) as in Section "Uncertainty analysis".

### Cloud water path change attribution

To diagnose the effect of increasing ice and liquid water paths, we use the binned approach of[65]. Liquid and ice water path is resolved into a histogram $n_j(W_k)$. Driving by each water path bin $W_k$ in each cloud phase $j$ is calculated separately according to

$$\frac{\partial F}{\partial T}\Big|_W = \sum_{j,k} f_j CRE_j(W_k)\frac{\partial n_j(W_k)}{\partial T}, \tag{7}$$

where $W$ is ice or liquid water path, $W_k$ is the water path bin, and $CRE_j(W_k)$ is the average observed cloud radiative effect for phase $j$ and water path bin $W_k$. Cloud radiative effect is diagnosed from observations by setting cloud fraction to zero and recalculating radiation with RRTM-LW. The term $\partial n_j(W_k)/\partial T$ is evaluated by regressing monthly-mean anomalies in the frequency of each water path bin $n_j(W_k)$ against monthly-mean anomalies in $T$. The cloud water path driver is the sum of the effects of increasing ice and liquid water paths. Uncertainty is calculated from (7) as in Section "Uncertainty analysis". By evaluating Eqs. (4), (6), and (7), we find the downward surface longwave radiation increase attributable to 8 drivers separated by cloud phase.

### Data availability
The data shown in the main text and visualized in figures is deposited on Zenodo (https://doi.org/10.5281/zenodo.15786066). The observational data used in this study[54,56–58,60,66,67] are available at the ARM Data Center (https://adc.arm.gov).

## Code availability

The radiative transfer code used for analysis is available at https://github.com/AER-RC/RRTM_LW. The code used to make the figures shown in this paper is available on Zenodo (https://doi.org/10.5281/zenodo.15786066).

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

## Acknowledgements

This work was supported by the NASA PREFIRE Mission (Award 849K995, JEK and LB), by the US Department of Energy (DOE) Atmospheric System Research (ASR) program (Grant no. DE-SC0013306, LB and GdB), and by an award to LB from the CIRES Graduate Student Research Award Program. GdB was supported in part by the NOAA Physical Sciences Laboratory. This work utilized the Alpine high-performance-computing resource at the University of Colorado Boulder. Alpine is jointly funded by the University of Colorado Boulder, the University of Colorado Anschutz, Colorado State University, and the National Science Foundation (Award 2201538). The authors would like to thank Dr. Matthew Shupe of the University of Colorado for helpful discussions of remote sensing instrumentation and retrievals at ARM NSA, as well as Dr. Jonah Shaw of the University of Colorado for helpful discussions on radiation attribution methods.

## Author contributions

L.B., J.E.K., and G.d.B. designed the study. L.B. derived the methods and performed the analysis. L.B. wrote the manuscript with contributions from J.E.K. and G.d.B. All authors contributed to interpretation of results.

## Competing interests

The authors declare no competing interests.
