## [Transparent Peer Review file · Nature Communications]

Increasing wintertime cloud opacity increases surface longwave radiation at a long-term Arctic observatory

Corresponding Author: Ms Leah Bertrand

Version 0:

Reviewer comments:

Reviewer #1

(Remarks to the Author)

In general, this is a well-written article, with solid scientific logic and the application of shared methods, which promises to understand if and how clouds can change the radiative budget at the surface in light of rising temperatures.

There are some aspects to work on, however. Two main general comments are provided here below. Some specific points are mentioned thereafter. I would sharpen the scientific reasoning in two spots: (1) the assumption of a steady surface, which is not warranted at all and (2) the data treatment. These are not minor changes to be underestimated.

General comments.

First, I don't believe that the results obtained from one site can be generalized to the entire Arctic. It is now common consensus in scientific literature that the Arctic climate is regionalized according to the type of surface: land ice (i.e. Greenland), marginal ice zone (from the Barents Sea to the Beaufort Sea, in an anti-clockwise direction), perennial sea ice (i.e. Central Arctic). So, the authors should refrain from the claim that their results apply to the full Arctic and specify throughout the text that the results hold only for a specific site and Arctic climates similar to Alaska.

Second, I found that the list of references was rather dated. Apart from two or three specific articles, much of the literature was published at least ten years ago. The authors should have put more effort into giving the reader a more comprehensive introductory overview of recent measurement campaigns in the Arctic and other sectors of the Arctic (e.g., MOSAIC, Ny Ålesund, airborne campaigns and so on). It's not just about the length of the measurement series; it's also about understanding regional differences between Arctic sectors.

Serreze's realization that the Arctic is divided into sectors and that the response of the Arctic climate depends on two main factors is well-founded: the surface area and the channels through which the Arctic exchanges air masses with the low latitudes. I list below in the specific comments some papers they want to dig in for relevant recent literature extraction.

Specific comments.

P2 L67-68: "... but they are rarely used to study cloud feedbacks directly because of their typically short-term records."

Not only because of this. The Arctic climate is highly regional. A single ground-based station cannot represent the Arctic condition in its entirety. Especially NSA, on land, and missing most of the sea-ice influences. So, modify the sentence to include the sparseness of the spatial sampling being one of the reason because they are rarely used.

P2 L85-86: "because it has the greatest surface warming without surface melt in the observational record of 1997 to 2024." Reference, please.

P2 L85-91: The train of thoughts of this paragraph is not precise and is a source of confusion. What do the authors mean stating that surface does not melt? Are you sure? And by what mechanisms? Are you referring to sea ice concentration, thickness, surface layer melting (ice-to-liquid), land ice?

Moreover, if clouds are going to reradiate downward and LW increases, does it not contribute to a warming, thus to an enhanced probability of surface melt?

And if the - likely glaciated - surface melts also due to temperature increase, does it not turn into snow? Does not snow increasingly insulate and suppress upwelling LW radiation, thereby invalidating your logic of assuming the surface signal steady during winter months?

P3 L95-98: "The warming effect of clouds increases to such a large extent that it overpowers the increased cooling effect by warming alone. In other words, cloud changes determine that more—rather than less—energy is trapped at the surface with warming."

For the sake of clarity, prepend "radiative" to cooling and remove "effect". It sounds odd.

P4 L166-169. This is clearly not a new result. It has been shown already for Svalbard and reproduced with the ICON model. See

Ebell, K., T. Nomokonova, M. Maturilli, and C. Ritter, 2020: Radiative Effect of Clouds at Ny-Ålesund, Svalbard, as Inferred from Ground-Based Remote Sensing Observations. *J. Appl. Meteor. Climatol.*, 59, 3–22, <https://doi.org/10.1175/JAMC-D-19-0080.1>.

and references herein (Fig 8, page 226)

Wendisch, M., and Coauthors, 2023: Atmospheric and Surface Processes, and Feedback Mechanisms Determining Arctic Amplification: A Review of First Results and Prospects of the (AC)3 Project. *Bull. Amer. Meteor. Soc.*, 104, E208–E242, <https://doi.org/10.1175/BAMS-D-21-0218.1>.

Of course, being here in winter time, no albedo from the surface is measurable, this then defaults to Fig. 6 in Ebell et al. 2020.

Other papers to consider, beside the MOSAIC general paper, are:

Maturilli, M., Herber, A. & König-Langlo, G. Surface radiation climatology for Ny-Ålesund, Svalbard (78.9° N), basic observations for trend detection. *Theor Appl Climatol* 120, 331–339 (2015). <https://doi.org/10.1007/s00704-014-1173-4>

Wendisch, M., and Coauthors, 2019: The Arctic Cloud Puzzle: Using ALOUD/PASCAL Multiplatform Observations to Unravel the Role of Clouds and Aerosol Particles in Arctic Amplification. *Bull. Amer. Meteor. Soc.*, 100, 841–871, <https://doi.org/10.1175/BAMS-D-18-0072.1>.

P5 L212-213: As per my comment above, are you able to conclusively rule out ice-to-snow melting so that any latent heat change can be excluded? Temperatures increase and moisture changes, especially in such a dry environment where the susceptibility is highest, are the drivers. I am not convinced that you can generalize your claim here.

P5 L223 and ff: Terminology used in this paragraph and ensuing sections. Is there a particular reason not to adopt customary wording like "Ice water path" and "Liquid water path"?

Using "cloud water path for ice-only" feels redundant and not streamlined. It puts a strain on the readers' focus. Also, why not using the SEB (Surface Energy Budget) acronym instead of "net surface longwave flux"?

P7 L300: Why are you citing Cox et al 2016? They look at springtime clouds, not winter clouds as in this study.

P7 L312-313: "In other words, net surface flux change is generally independent of the circulation."

Two remarks:

1) Are clouds dependent on circulation? If they do, is not this an indirect effect of atmospheric circulation on SEB?

2) There is quite a body of literature on this topic. Could not be that your multi-decadal component in the record overshadows shorter oscillations? This being the case, the inference of your regression is not scientifically conclusive.

P9 L411-413: "The exception is underestimation from 2004 to 2006, but removing this data has no effect on our conclusions (not shown)."

Why? This is source of perplexity. Whenever I find outliers in my results, I put efforts in explaining and not skipping them. I do not understand the logic of the authors in ad-hoc removing data from the batch that do not fit the picture.

What are the causes of exceeding the observational uncertainty? the authors shall go to the bottom of it because either is the observational record or the closure. Both affect the numbers of the paper. Tertium non datur.

P10 L433-434: "Consistent performance over time shows that the data are suitable for long-term analysis."

When not accounting for years 2004-2006, right?

Reviewer #2

(Remarks to the Author)

- What are the noteworthy results?

The paper shows a long-term trend in cloud properties with implications for the TOA radiation over the Arctic. This is important because it signifies sustained global change. This paper leverages surface observations with a long data record to provide information about the long-term climactic evolution of this region and uses model output from RRTM to further explain this evolution.

- Will the work be of significance to the field and related fields? How does it compare to the established literature? If the work is not original, please provide relevant references.

The work is original and very timely. It makes a ton of sense and is a clever way of leveraging the data record – it is refreshing to see surface observations used in these kinds of studies. There is nothing overly exotic in the methodology that should raise concern about the robustness of the results.

I note that the paper is very nicely written and easy to follow. While it shouldn't impact the editorial decision, I note that the corresponding author is a graduate student with one other publication. Really impressive!

- Does the work support the conclusions and claims, or is additional evidence needed?

Yes. The use of a closure in radiation is nice and makes the results robust.

- Are there any flaws in the data analysis, interpretation and conclusions? Do these prohibit publication or require revision?

While this is a strong work, and the findings are likely robust, the authors imply (most clearly in the beginning of Section 5) that the results presented are in relation to “the Arctic”. However, this paper is purely based on analysis from just one surface site: NSA. Can we consider NSA to be representative of the Arctic as a whole in this study? Could there be local variability in decadal temperature/radiation change at NSA that is not present elsewhere in the Arctic, skewing the results away from a regional mean?

Is the methodology sound? Does the work meet the expected standards in your field?

The methodology is sensible, easy-to-follow, and smartly leverages the ARM data to perform long-term analysis. This is exciting since the data from ARM is getting to the length that studies like this are possible.

Figure 1: the trends would be a bit easier to interpret visually if ranges on the best fit line were added.

Figure 2e: These colors of blue are hard to distinguish.

Section 4- the discrimination by SLP is cool, and it makes sense that the authors want to rule out synoptic state. Are there trends in SLP? In line 319 the statement that these trends are not due to circulation is made. This feels like it could be supported a bit more rigorously given the strength of this statement by showing that the trend is not explainable by shifting in SLP across the decades between different bins in Fig. 4.

- Is there enough detail provided in the methods for the work to be reproduced?

Line 114: A few notes about the observations from NSA being used would be helpful. Obviously these publications are very limited in length, but a sentence here describing the specific observation/product would help with reader comprehension. There is a lot of detail under section 6, but given that ARM data is not frequently used for these kinds of studies it would be helpful to have a few sentences in the main text characterizing the observations being used and then the reader can dig in in the methods.

Reviewer #3

(Remarks to the Author)

Reviewer #4

(Remarks to the Author)

This paper presents a comprehensive analysis of the winter surface radiation budget and how this has evolved as a consequence of Arctic climate warming, specifically how clouds have changed to affect this. To my mind represent the first

time this has been analyzed using direct measurements, not inferred from satellite retrievals (which are observations but indirect) and not from modeling – which is iffy in the Arctic where models struggle to represent even the current climate processes let alone how they change. The results are very interesting and the arguments appears solid; to me it was a surprise that the largest effect came from more opaque ice-only clouds. I think the current wisdom has more focused on increasing fraction of liquid in mixed-phase clouds and hence these have been a focus for cloud physics research; this study indicate that a larger focus should be on ice clouds. These distinctions are important to get right in models, especially since many models tends to overestimate either liquid or ice. The argument is that it is the ice clouds that holds potential to become more optically thick, while clouds containing liquid would to a larger extent already behave like black bodies while the changes in the occurrence of liquid-bearing clouds does not compensate for this, seems logical and in my mind is the take-home message of the paper, besides the fact that clouds do matter for the Arctic amplified warming.

However, the paper is not particularly well organized nor well written, making it hard to follow the arguments and some details are missed all together, like the fact that some of the attribution rests on radiative transfer modelling and not on the observations; I didn't get that until I read the methods section. It is not that I object; more that I would have wanted to know. I think the manuscript needs a real work-over before it is published and therefore I recommend major revision.

Major comments:

I think the author needs to think more about the narrative. A paper is not just the results and the methods; it needs to tell a story and it should be a story that is easy to read and grasp as far as is possible. The main problem I had with this paper is not understanding how the different classifications were accomplished. I did get that when I read the methods section but by the the damage was already done. The order of presentation also matters; start with a hypothesis (why), then a brief description of what the classes are (how) and then the results. In this paper there are many places where a section is started with the conclusions followed by the results and then discussion but there is no background; no hypothesis and no "why". This paper also has a lot of unnecessary repetitions where something is stated, then shown how, and then the same thing is restated again at the end of the same argument. There is very little background; for example what the different feedback mechanisms are or what we would expect to see; there are no hypotheses and to understand how the conclusions were drawn one have to wait until the methods section.

The record lengths are skipped over rather carelessly on page two; over two decades of radiation measurements and a decade of cloud observations. But the cloud observations are crucial for the conclusions in the paper and those are only – what – 10 years? So exactly how long records are used for what parts of the results? This needs to be up front and center; I have no problems with the trend analyses of the radiation data, but the conclusions rest on 10 years of cloud data; barely enough to define the state of things let alone a trend.

There is also a lot of discussion of significant and non-significant trends or differences, but I could not find anything about how the significance was tested.

Minor comments:

Line 38: Here increased opacity of ice-only and mixed phase clouds contribute equally, but that is not what it looks like in the text; there it seems that ice-only clouds have a larger contribution.

L 40: What do you mean by "sets the sign of"?

L 50: If "trapping" of heat below the clouds was the process here, how could that not show up at TOA and be measured by satellite? If heat is trapped, there has to be less escaping to space?

L 55-53: Correct me if I'm wrong, but the warming of the Arctic must be a consequence of the mismatch between heat advected into the region from south and changes in the TOA. That defines climate; not what happens at the surface. So why should factors contributing to climate change not use TOA? If clouds in the Arctic were to warm the surface more, then this begs the question from where comes that energy? The problem with passive sensor satellites is that they needs retrievals that are notoriously unreliable especially in the Arctic; not that they measure the wrong thing. Moreover, many of the problems with active sensor satellites are mostly due to design choices – they miss most of the low clouds which happens to be dominant in the Arctic, and they just barely reach into the souther parts of the Arctic.

L 74-77: The fact that others have used short records is a poor excuse; did you look into the effects of the short record? Also what record length is used for the cloud data?

Section starting at line 134: I would like to see an expanded discussion here, especially of how changes in cloud phase, cloud fractions and temperature would affect the local minimum in the distribution of net radiation. Is the same value used to distinguish clear and cloudy even when temperature and clouds change? On L 176-177 you state they do not change. I wonder why and is that evident?

Line 213: What about sensible heat?

L 217: Just a reference forward in the text doesn't hack it; you need some description of it here for the results to mean anything. No one is going jump forward and read the methods section before reading the rest of the text first.

L 220: What do you mean by "only explains 76%"; that seems like a lot to me?

L 228-229: How "suppress"? What do you mean by this?

L 254-258: This is mostly a repetition to things already stated.

L 260-263: The second of these sentences seems like a repletion of the first.

Section 4: Is surface pressure really a good metric for circulation variability. Think about a cyclone passing northward to your east or to your west. The pressure could be identical and yet one would represent advection of dry and cold Arctic air while the other would bring moist warm mid-latitude air. This section is not very convincing.

L 393-394: Are soundings also included here?

Version 1:

Reviewer comments:

Reviewer #1

(Remarks to the Author)

The authors have done a considerable job in addressing my remarks. As far as I am concerned, the paper can be published as it is.

Still, at authors' discretion, a closing speculative sentence in the discussion section on future steps is welcome.

To be precise: according to the authors, what are the current limitations of their study and what can be done to extend the presented results both in terms of time (other seasons) and in terms of methodology and data? In fact, a concluding message to the community on what the authors believe is necessary to further explore the topic.

(Remarks on code availability)

Reviewer #2

(Remarks to the Author)

I thank the authors for their time thoroughly responding to my questions. I have no further suggestions or questions at this time.

(Remarks on code availability)

There appears to be a readme file with install instructions.

I appreciate that reproducible code is important and that the journal is trying to insure transparency, but installing and running the code is beyond the amount of time I can really devote to this review.

Reviewer #3

(Remarks to the Author)

(Remarks on code availability)

Reviewer #4

(Remarks to the Author)

This is a revised version of a study that – with the exception of using a radiative transfer code – uses ground-based observations to draw conclusions on cloud feedbacks on surface energy fluxes in the Arctic. If my previous review came across as critical, let me start here by saying that I find this work ingenious and novel and with some important, to me somewhat surprising, conclusions; it should be published. At this time I also want to congratulate the authors on the revision, that has made the manuscript much more enjoyable to read. The reservations I had for it lacking a narrative are alleviated. Also on the quite a few detailed comments, the authors have made a thorough job. Although we are not in complete agreement on all items, I will write this down as a matter of opinion.

Hence, it is my recommendation at this time is that it be published after a minor revision. I have, however, a few remaining comments that I would wish the authors to consider:

I find the use of the term “phase” somewhat confusing and I would recommend that the authors use “cloud type” rather than “cloud phase” whenever they discuss the three main types of clouds: “ice-only”, “mixed-phase” and “liquid-only”. They should use the term “phase” only when discussing the effects of changes in phase. This is anyway how I interpret the results presented in Figure 2, and the discussion relating to it; that the 3rd column from the right in Figure 2e refers to effects of changes in cloud phase, i.e. ice clouds more often becoming mixed-phase or liquid only with the warming. As the discussion turns to Figure 3, the word “phase” now seem to change meaning to “type”; at least that is how I understand the caption to Figure 3, where “+Clouds” now mean all the 4 effects for the different “types” (= “phases”?). After all, cloud water only has two phases; liquid or ice. And vapor of course, but then its not a cloud anymore´...

On Lines 348-350:

Still, the three lowest-pressure bins has a significantly larger effects than the three highest-pressure bins. Also, I would not use the word “confound” here; you find that the temperature anomalies better explain the cloud effects than the pressure anomalies, but the temperature anomalies themselves comes from somewhere, possibly from changes in large scale advection. So maybe the local effects are temperature driven but the temperature changes are – in part – “weather driven”. Line 366-368: I still struggle with the conceptualization of climate and climate change here. Bottom line, climate change in a

region is due to a mismatch between net radiation at TOA and import of heat over the boundaries. Now you say that temperature increase may be additionally enhanced by local changes, in clouds, a radiatively driven additional warming (“accumulation of surface energy”) near the surface from changing clouds that is not detectable from space? So where does that energy come from, if not from advection?

On Lines 463-473: While I’m not necessarily questioning the use of RRTM-LW as such, and I’m not worried by the comparison statistics between modelled and observed radiation either; its well within the measurement uncertainty. However, I’m still curious; isn’t there a consistent (except the very last years) negative bias, or maybe even a positive trend in bias? And what happened with the missing years in Figure 5a? What is the reason for the distribution in Figure 5b being the “fattest” for mid-range values, around 180 Wm^{-2} ? Are those cases with a broken cloud field? And how do you deal with that in the model; care to comment?

(Remarks on code availability)

The images or other third party material in this Peer Review File are included in the article’s Creative Commons license, unless indicated otherwise in a credit line to the material. If material is not included in the article’s Creative Commons license and your intended use is not permitted by statutory regulation or exceeds the permitted use, you will need to obtain permission directly from the copyright holder.

**Dear Reviewers,**

We thank you all for your thoughtful and constructive comments and suggestions. Your reviews have
greatly benefitted the science in our manuscript.

We highlight some key improvements here:

- 1) We revised our article to no longer generalize the North Slope of Alaska observatory to the
entire Arctic.
- 2) We revised our article to improve the precision, structure, and flow of the text.
- 3) We expanded citations and reference to previous literature.
- 4) We clarified and expanded our analysis methods. Specifically, we added a dedicated description
of our statistical significance/uncertainty propagation procedure. We also improved our
radiative closure description placing it in context of previous similar work. We also performed
additional analysis wherever we could to address reviewer concerns/questions about the
circulation confounder analysis (lines 441-462 and 741-778 of this document), surface melt
(lines 122-138 of this document), and independence of surface and top-of-atmosphere radiation
(lines 607-622 of this document). Our response to reviewers also includes new analyses
requested by the reviewers. These analyses have been added to a supplement and referenced in
the main text.

Our revised manuscript is ready for review. Thank you for your time.

Best,
Leah Bertrand, Jennifer E. Kay, and Gijs de Boer

**Reviewer #1 (Remarks to the Author)** 29

In general, this is a well-written article, with solid scientific logic and the application of shared methods,
which promises to understand if and how clouds can change the radiative budget at the surface in light
of rising temperatures.

There are some aspects to work on, however. Two main general comments are provided here below.
Some specific points are mentioned thereafter. I would sharpen the scientific reasoning in two spots: (1)
the assumption of a steady surface, which is not warranted at all and (2) the data treatment. These are
not minor changes to be underestimated.

**Response:** We thank Reviewer #1 for their thorough, constructive, and largely positive review.

**General comments.**

First, I don't believe that the results obtained from one site can be generalized to the entire Arctic. It is
now common consensus in scientific literature that the Arctic climate is regionalized according to the
type of surface: land ice (i.e. Greenland), marginal ice zone (from the Barents Sea to the Beaufort Sea, in
an anti-clockwise direction), perennial sea ice (i.e. Central Arctic). So, the authors should refrain from

the claim that their results apply to the full Arctic and specify throughout the text that the results hold
only for a specific site and Arctic climates similar to Alaska.

**Response:** We agree with the reviewer that results from one site should not be generalized to the entire
Arctic. We thank the reviewer for the push to be more precise about the area over which our findings
are valid. In response, we now use “North Slope of Alaska (NSA)” instead of the “Arctic” throughout the
text (see lines 41, 268, 319, 368, and 406 of the revised manuscript).

Second, I found that the list of references was rather dated. Apart from two or three specific articles,
much of the literature was published at least ten years ago. The authors should have put more effort
into giving the reader a more comprehensive introductory overview of recent measurement campaigns
in the Arctic and other sectors of the Arctic (e.g., MOSAIC, Ny Ålesund, airborne campaigns and so on).
It's not just about the length of the measurement series; it's also about understanding regional
differences between Arctic sectors.

Serreze's realization that the Arctic is divided into sectors and that the response of the Arctic climate
depends on two main factors is well-founded: the surface area and the channels through which the
Arctic exchanges air masses with the low latitudes. I list below in the specific comments some papers
they want to dig in for relevant recent literature extraction.

**Response:** We cite the original papers that provide essential framing and context for our work. These
papers were the first present the bi-modal structure of longwave radiation (Stramler et al. 2011,
Morrison et al. 2012). While “dated”, we think these studies are essential to cite as they are the first to
discover the phenomena that we address here.

Recent field campaigns, while exciting, have not transformed or challenged the ideas in these original
papers. Therefore, we think providing a “more comprehensive introductory overview of recent
measurement campaigns” is not strictly needed. This is not a review paper. That said, we did add several
recent studies to the list of field campaigns that observed wintertime net longwave radiation bimodality
on line 141 including: Graham et al., 2017 (Atlantic Arctic sector winter); Raddatz et al., 2015 (Canadian
Arctic winter); Solomon et al., 2023 (MOSAIC winter); Yamanouchi, 2019 (Ny Ålesund winter).

Specific comments.

P2 L67-68: "... but they are rarely used to study cloud feedbacks directly because of their typically short-
term records."

Not only because of this. The Arctic climate is highly regional. A single ground-based station cannot
represent the Arctic condition in its entirety. Especially NSA, on land, and missing most of the sea-ice
influences. So, modify the sentence to include the sparseness of the spatial sampling being one of the
reason because they are rarely used.

**Response:** We agree that a single site is not representative of the entire Arctic. See our response on
lines 50-53 of this response to reviewer in which we describe how we changed from using “Arctic” to
using “North Slope of Alaska (NSA)”.

As NSA is less than 2 km away from the coastal boundary, and has frequent wintertime northerly winds
(Mülmenstädt et al., 2012 Fig. 2), measurements at NSA are affected by both sea ice/ocean and land.

The contrast between the land and ocean/sea ice is likely maximum when the sea ice is thinnest/not
present and when the land is heated more than the ocean. During winter, both the land and the sea ice
are covered by snow. Thus, winter is likely the season when the difference between the land and ocean
influences is the smallest.

For example, DJFM during the SHEBA field campaign had sensible heat flux of about -5 W/m^2 and latent
heat of about $<\pm 1 \text{ LH W/m}^2$ (Persson, 2002). ARM NSA, by comparison, has DJFM average sensible heat
flux of -7.2 W/m^2 and latent heat -0.2 W/m^2 . We have clarified our argument for ARM NSA's relevance
on lines 87-100 of the manuscript.

P2 L85-86: "because it has the greatest surface warming without surface melt in the observational
record of 1997 to 2024." Reference, please.

**Response:** We do not provide a reference because we can support this claim by our analysis of ARM NSA
observations. Please see Figure 1 in the following comment showing that near-surface air temperatures
are inconsistent with the reviewer's suggestion of possibly melting snow at our coastal land site.

P2 L85-91: The train of thoughts of this paragraph is not precise and is a source of confusion. What do
the authors mean stating that surface does not melt? Are you sure? And by what mechanisms? Are you
referring to sea ice concentration, thickness, surface layer melting (ice-to-liquid), land ice?

Moreover, if clouds are going to reradiate downward and LW increases, does it not contribute to a
warming, thus to an enhanced probability of surface melt?

And if the - likely glaciated - surface melts also due to temperature increase, does it not turn into snow?
Does not snow increasingly insulate and suppress upwelling LW radiation, thereby invalidating your logic
of assuming the surface signal steady during winter months?

**Response:** Near-surface air temperatures are inconsistent with surface snowmelt during winter at ARM
NSA (Figure 1 below). We investigated near-surface air temperatures and found that 99.9% of
observations were below -0.5°C . We have clarified this point in the text by revising lines 87-89 to read
"Our analysis is limited to winter alone (December-March) because it has the greatest surface warming
without surface melt (99.9% of near-surface air temperature observations $<-0.5^\circ\text{C}$)." Thus, surface melt
is negligible at this snow-covered coastal land site (surrounded by pack ice) during winter.

The reviewer is right to point out that changes to other terms in the surface energy budget (sensible and
latent heat, subsurface heat flux) could suppress the upwelling longwave response to warming.

However, the observed upwelling longwave response to warming provides no evidence for such a
suppression. The observed increase agrees completely with the increase expected from a Planck

response alone (Figure 2 below). We have included this additional analysis in the supplement to the
manuscript (Supplementary Note 1, Supplementary Figure 2, Supplementary Figure 3).

Response to Reviewers Figure 1. Minute-frequency cumulative distribution function of December-March near-surface air temperature at ARM NSA, 1998-2023.

Figure 2. Planck expectation for upwelling longwave increase with warming from monthly-mean surface skin temperatures (black lines) vs. observed response (red lines). Surface skin temperatures derived from observed upwelling longwave radiative fluxes assuming an emissivity of 1). 95% confidence intervals indicated via dashed lines.

P3 L95-98: "The warming effect of clouds increases to such a large extent that it overpowers the increased cooling effect by warming alone. In other words, cloud changes determine that more—rather than less—energy is trapped at the surface with warming."

For the sake of clarity, prepend "radiative" to cooling and remove "effect". It sounds odd.

Response: We thank the reviewer for the suggestion to improve the clarity of this sentence. The sentence now reads "The warming effect of clouds increases to such a large extent that it overpowers the increased radiative cooling implied by warming alone."

P4 L166-169. This is clearly not a new result. It has been shown already for Svalbard and reproduced with the ICON model. See

Ebell, K., T. Nomokonova, M. Maturilli, and C. Ritter, 2020: Radiative Effect of Clouds at Ny-Ålesund,
Svalbard, as Inferred from Ground-Based Remote Sensing Observations. *J. Appl. Meteor. Climatol.*, 59,
3–22, <https://doi.org/10.1175/JAMC-D-19-0080.1>.

and references herein (Fig 8, page 226)

Wendisch, M., and Coauthors, 2023: Atmospheric and Surface Processes, and Feedback Mechanisms
Determining Arctic Amplification: A Review of First Results and Prospects of the (AC)3 Project. *Bull.*
*Amer. Meteor. Soc.*, 104, E208–E242, <https://doi.org/10.1175/BAMS-D-21-0218.1>.

Of course, being here in winter time, no albedo from the surface is measurable, this then defaults to Fig.
6 in Ebell et al. 2020.

Other papers to consider, beside the MOSAIC general paper, are:

Maturilli, M., Herber, A. & König-Langlo, G. Surface radiation climatology for Ny-Ålesund, Svalbard (78.9°
176 N), basic observations for trend detection. *Theor Appl Climatol* 120, 331–339
(2015). <https://doi.org/10.1007/s00704-014-1173-4>

Wendisch, M., and Coauthors, 2019: The Arctic Cloud Puzzle: Using ACLOUD/PASCAL Multiplatform
Observations to Unravel the Role of Clouds and Aerosol Particles in Arctic Amplification. *Bull. Amer.*
*Meteor. Soc.*, 100, 841–871, <https://doi.org/10.1175/BAMS-D-18-0072.1>.

**Response:** From the reviewer text, we unclear about what is “clearly not a new result.” Two claims are
made in L166-196 of the original manuscript:

Claim (1): “observations show a shift from the clear to the opaque mode with warming (Fig. 1c,d)” and
Claim (2): “Clouds are implicated in this change because warming and increasing greenhouse gases
alone (Methods, Sec. 6.3) have no effect on opaque mode frequency (Fig. 1c, dashed line).”

Below we review why our text is appropriate as stated for both claims and why we did not change it.
Instead, we added the following to lines 141-143: “Some studies have used indirect evidence to suggest
that opaque mode frequency may increase with warming in the Atlantic sector of the Arctic (Graham et
al., 2017; Nomokonova et al., 2020), but this has never been directly observed.”

For Claim (1) – we searched the literature starting from the reviewer’s suggested citations and did not
find any study prior to ours reporting this result. Specifically, we could not find any other studies
analyzing long-term ground-based observational datasets and reporting increasing wintertime net
surface longwave in response to warming. See our literature review below for a list of sources we
examined and their relevance to our analysis.

The main study the reviewer recommended (Ebell et al. 2020) only looks at three years of radiative
fluxes and does not report any radiative response to warming. Individual field campaigns including those
suggested by the reviewer as a part of the AC3 project or MOSAIC are not able to detect such a long-
term longwave radiation change due to their limited duration.

Claim (2) is well-known from the early 1998 SHEBA field campaign, which is already cited in our paper
(Morrison et al., 2012; Stramler et al., 2011). We do not claim it is a new result, therefore we do not
investigate it further.

We review the literature suggested by the reviewer (in blue) and indicate their relevance (in red) to
claim (1) of a shift from the clear to opaque radiative mode with warming in what follows.

Ebell, K., T. Nomokonova, M. Maturilli, and C. Ritter, 2020: Radiative Effect of Clouds at Ny-Ålesund,
Svalbard, as Inferred from Ground-Based Remote Sensing Observations. *J. Appl. Meteor. Climatol.*, 59,
3–22, <https://doi.org/10.1175/JAMC-D-19-0080.1>.

This paper only has three years of surface radiative fluxes. It does not quantify the response to warming.

Wendisch, M., and Coauthors, 2023: Atmospheric and Surface Processes, and Feedback Mechanisms
Determining Arctic Amplification: A Review of First Results and Prospects of the (AC) 3 Project. *Bull.*
*Amer. Meteor. Soc.*, 104, E208–E242, <https://doi.org/10.1175/BAMS-D-21-0218.1>.

The most relevant passage from Wendisch et al. 2023 is “Ground-based observations at the AWIPEV
research base hint at an increase of cloud occurrence, liquid water path (LWP), and ice water path (IWP)
in all seasons (Nomokonova et al. 2020). Using a new retrieval algorithm (Nakoudi et al. 2021b), the
long-term analysis (2011–20) of data collected at AWIPEV revealed that in winter and spring cirrus
clouds are thicker and seem to appear more frequently (Nakoudi et al. 2021a). From the preliminary
analysis of satellite data from previous research outside (AC)³, it appears that an increase in cloud LWP
has resulted in a positive trend in COT for the liquid phase of 2.8% decade⁻¹ and a negative trend in COT
for the ice phase of -6.1% decade⁻¹ (Lelli et al. 2022).” Thus we add Nomokonova et al. (2020), Nakoudi
et al. (2021), and Lelli et al. (2023) to the list and discuss them below. We note that Wendisch et al. 2023
states that observations “hint” at an increase in cloud occurrence, LWP, and IWP based on 2-3 years of
surface-based cloud retrievals.

Of course, being here in winter time, no albedo from the surface is measurable, this then defaults to Fig.
6 in Ebell et al. 2020.

Other papers to consider, beside the MOSAIC general paper, are: Shupe, Matthew D., et al. "Overview of
the MOSAIC expedition: Atmosphere." *Elem Sci Anth* 10.1 (2022): 00060.

This overview paper does not plot distributions of surface net longwave radiation. But we cited a paper
based on MOSAIC data in the revised manuscript which does investigate the distribution of net
longwave radiation (Solomon et al., 2023).

Maturilli, M., Herber, A. & König-Langlo, G. Surface radiation climatology for Ny-Ålesund, Svalbard (78.9°
244 N), basic observations for trend detection. *Theor Appl Climatol* 120, 331–339
(2015). <https://doi.org/10.1007/s00704-014-1173-4>

This paper reports that surface net longwave radiation during winter and spring is not confidently
increasing. Winter is 3.9±3.9 W/m²/decade. They do not investigate the distribution of radiation.

Wendisch, M., and Coauthors, 2019: The Arctic Cloud Puzzle: Using ACLOUD/PASCAL Multiplatform
Observations to Unravel the Role of Clouds and Aerosol Particles in Arctic Amplification. *Bull. Amer.*

Meteor. Soc., 100, 841–871, <https://doi.org/10.1175/BAMS-D-18-0072.1>.

This is a short <1 year field campaign, hence it cannot investigate the response to warming.

Our comments on the secondary literature contained in the reviewer’s suggestions:

Nomokonova, Tatiana, et al. "The influence of water vapor anomalies on clouds and their radiative
effect at Ny-Ålesund." *Atmospheric Chemistry and Physics* 20.8 (2020): 5157-5173.

This paper only investigates 2.5 years of cloud property measurements. Further, surface net longwave is
never discussed in this paper. However, it has some similarities to our study, so we mention it in line 143
the revised manuscript.

Lelli, Luca, et al. "Satellite remote sensing of regional and seasonal Arctic cooling showing a multi-
decadal trend towards brighter and more liquid clouds." *Atmospheric Chemistry and Physics* 23.4 (2023):
2579-2611.

This paper is satellite passive remote sensing instead of in-situ data. Thus, it is not a reliable record of
the Arctic surface energy budget.

Nakoudi, K., Ritter, C., & Stachlewska, I. S. (2021). Properties of Cirrus Clouds over the European Arctic
(Ny-Ålesund, Svalbard). *Remote Sensing*, 13(22), 4555. <https://doi.org/10.3390/rs13224555>.

This paper does not discuss the surface energy budget. They found no trends found in cloud properties:
“Over the 10 years of the analysis, the cirrus properties have not exhibited any clear temporal trend.”
Additionally, no relationship was found between cloud opacity and temperature (Figure 10).

P5 L212-213: As per my comment above, are you able to conclusively rule out ice-to-snow melting so
that any latent heat change can be excluded? Temperatures increase and moisture changes, especially
in such a dry environment where the susceptibility is highest, are the drivers. I am not convinced that
you can generalize your claim here.

**Response:** We rule out latent heat exchange based on near-surface air temperatures (see Fig. 1 above).

P5 L223 and ff: Terminology used in this paragraph and enqing sections. Is there a particular reason not
to adopt customary wording like "Ice water path" and "Liquid water path"?

Using "cloud water path for ice-only" feels redundant and not streamlined. It puts a strain on the
readers' focus.

**Response:** We do not adopt “Ice water path” and “Liquid water path” as the reviewer suggests because
these terms do not indicate overall cloud phase. If we did adopt these terms, we would need to use
statements like “ice water path in mixed-phase clouds” or “ice water path in ice-only clouds”. Therefore,
we kept the text as is using “cloud water path” to reduce the overall number of terms.

Also, why not using the SEB (Surface Energy Budget) acronym instead of "net surface longwave flux"?

**Response:** We use “Net surface longwave flux” because it is accurate. In contrast, using SEB is not
accurate because SEB includes turbulent and conductive heat fluxes which we do not analyze.

P7 L300: Why are you citing Cox et al 2016? They look at springtime clouds, not winter clouds as in this
study.

**Response:** We thank the reviewer for pointing this out. We removed citation of Cox et al. 2016.

P7 L312-313: "In other words, net surface flux change is generally independent of the circulation."

Two remarks:

1) Are clouds dependent on circulation? If they do, is not this an indirect effect of atmospheric
circulation on SEB?

2) There is quite a body of literature on this topic. Could not be that your multi-decadal component in
the record overshadows shorter oscillations? This being the case, the inference of your regression is not
scientifically conclusive.

**Response:** We have already done analysis in the paper to show that that the longwave changes at NSA
are not an "indirect effect of atmospheric circulation". This point was described in the original paper at
lines 317-350. We answer the reviewer's questions below but given that this was already described: we
have not modified the original text.

*"1) Are clouds dependent on circulation?"*

**Response:** Yes. Clouds do depend on circulation and this dependence allows cloud properties to
influence longwave radiation.

*"If they do, is not this an indirect effect of atmospheric circulation on SEB?"*

**Response:** No. We find that the longwave radiation response to warming, and cloud opacity increase
occur during both high and low local surface pressures (Figure 4).

Also, we analyze longwave radiation, not SEB. See line 297-298 of this document.

*"2) There is quite a body of literature on this topic. Could not be that your multi-decadal component in
the record overshadows shorter oscillations? This being the case, the inference of your regression is not
scientifically conclusive."*

**Response:** We are unsure what literature the reviewer is referring to. Our analysis shows that the
dependence of longwave radiation on circulation-driven cloud variations does not explain our results.
Therefore we have not modified the original text.

We do not agree with the idea that a "multi-decadal component in the record overshadows shorter
oscillations." No variables in the analysis other than temperature had a secular trend. Local sea level
pressure, net surface longwave flux, and cloud water path did not have any significant trend over time
(see manuscript Fig. 1a and response to reviewers Fig. 3). We also investigated the effect of pressure
variations over multiple decades (lines 443-464 in this document) and found that it had no effect on net
surface longwave flux.

P9 L411-413: "The exception is underestimation from 2004 to 2006, but removing this data has no effect
on our conclusions (not shown)."

 Why? This is source of perplexity. Whenever I find outliers in my results, I put efforts in explaining and
 not skipping them. I do not understand the logic of the authors in ad-hoc removing data from the batch
 that do not fit the picture.
 What are the causes of exceeding the observational uncertainty? the authors shall go to the bottom of it
 because either is the observational record or the closure. Both affect the numbers of the paper. Tertium
 non datur.

 **Response:** First and importantly, we agree with the reviewer that ad-hoc removing of data is bad
 science. We have not done that here. We emphasize that results presented in the paper include all data
 358 years (2004-2019). In the original text, we simply noted that our conclusions are not dependent on the
 359 inclusion of data years 2004-2006. We have removed this statement in the revised manuscript. In the
 360 revised manuscript, we argue that our degree of closure for each year is consistent with existing state-
 361 of-the-art Arctic closure exercises.

 For reference, please compare our closure statistics for each year (Table 1) to those found in the
 literature (Table 2) for Arctic surface downwelling longwave radiation closure from ground-based
 measurements (Barrientos-Velasco et al., 2022, 2024; Ebell et al., 2020; Griesche et al., 2024; Shupe et
 al., 2015). No year in our study has statistics outside of the range of values reached in past studies.

Statistic (W/m ²)	2004	2005	2006	2008	2009	2010	2011	2014	2015	2016	2017	2018	2019
median	-3.91	-6.06	-5.79	-1.95	-3.50	-2.09	-1.79	-2.89	-2.03	-3.55	-1.27	0.09	1.00
IQR	10.06	14.92	18.80	9.38	10.66	11.51	13.86	9.18	14.92	10.42	17.69	12.07	16.18

Table 1. Closure statistics for each year of the present study

Citation	Study duration	Median (W/m ²)	IQR (W/m ²)
Present study	13 years	-2.6	13.1
Shupe et al. 2015	2 years	2.3	10.1
Ebell et al. 2020	3 years	0.7	12.2
Griesche et al. 2024	1 month	-3	8
Barrientos-Velasco et al. 2023	3 days	-19	Unknown
Barrientos-Velasco et al. 2024	1 year	3.2	18.8*

Table 2. Literature review of closure statistics for surface downwelling longwave radiative flux from
 Arctic surface-based observations.

* Converted from standard deviation assuming normal distribution (IQR = 1.34*stdev)

 P10 L433-434: "Consistent performance over time shows that the data are suitable for long-term
 analysis."

When not accounting for years 2004-2006, right?

As emphasized in the previous comment, performance over the entire data record is consistent with
other state-of-the-art Arctic radiative closure studies. See lines 356-373 of this response to reviewers.

Reviewer #2 (Remarks to the Author)

• What are the noteworthy results?

The paper shows a long-term trend in cloud properties with implications for the TOA radiation over the
Arctic. This is important because it signifies sustained global change. This paper leverages surface
observations with a long data record to provide information about the long-term climactic evolution of
this region and uses model output from RRTM to further explain this evolution.

• Will the work be of significance to the field and related fields? How does it compare to the established
literature? If the work is not original, please provide relevant references.

The work is original and very timely. It makes a ton of sense and is a clever way of leveraging the data
record – it is refreshing to see surface observations used in these kinds of studies. There is nothing
overly exotic in the methodology that should raise concern about the robustness of the results.

I note that the paper is very nicely written and easy to follow. While it shouldn't impact the
editorial decision, I note that the corresponding author is a graduate student with one other publication.
Really impressive!

• Does the work support the conclusions and claims, or is additional evidence needed?

Yes. The use of a closure in radiation is nice and makes the results robust.

• Are there any flaws in the data analysis, interpretation and conclusions? Do these prohibit publication
or require revision?

**Response:** We thank Reviewer #2 for their thorough, constructive, and largely positive review.

While this is a strong work, and the findings are likely robust, the authors imply (most clearly in the
beginning of Section 5) that the results presented are in relation to “the Arctic”. However, this paper is
purely based on analysis from just one surface site: NSA. Can we consider NSA to be representative of
the Arctic as a whole in this study? Could there be local variability in decadal temperature/radiation
change at NSA that is not present elsewhere in the Arctic, skewing the results away from a regional
mean?

**Response:** We agree. A similar point was also made by reviewer #1. See line 89 of this response to
reviewer document. Thus, we have changed from “the Arctic” to “North Slope of Alaska”.

Is the methodology sound? Does the work meet the expected standards in your field?

The methodology is sensible, easy-to-follow, and smartly leverages the ARM data to perform long-term
analysis. This is exciting since the data from ARM is getting to the length that studies like this are
possible.

Figure 1: the trends would be a bit easier to interpret visually if ranges on the best fit line were added.

**Response:** We agree. We added 95% confidence intervals of trendline fit.

Figure 2e: These colors of blue are hard to distinguish.

**Response:** We agree. We changed the shades of blue to be easier to distinguish.

Section 4- the discrimination by SLP is cool, and it makes sense that the authors want to rule out
synoptic state. Are there trends in SLP? In line 319 the statement that these trends are not due to
circulation is made. This feels like it could be supported a bit more rigorously given the strength of this
statement by showing that the trend is not explainable by shifting in SLP across the decades between
different bins in Fig. 4.

**Response:** We agree that assessing the change of SLP over time is useful. We did so and found SLP has
no trend over time (Figure 3). We also found that SLP variability over decades has no effect on net
radiation (lines 448-463). Since this additional analysis does not change our conclusions, we have not
modified our initial analysis. Instead, we added this additional figure to the supplement and referenced
it in the text on line 346.

We evaluated the effect of SLP variability on net longwave flux over decades in response to the
reviewer's suggestion to show "that the trend is not explainable by shifting in SLP across the decades
between different bins". The shifting SLP between the first and last decades of observations is shown in
the left panel in Figure 4. The dependence of net longwave flux on SLP is shown on the right panel in
Figure 4. The radiative effect of shifting SLP on net longwave is the inner product of these two
quantities: $\Delta F_N(\Delta SLP) = \sum_i \Delta f(SLP_i) F_N(SLP_i)$, where Δf is the change in SLP and $F_N(SLP_i)$ is the
average net flux anomaly within SLP bin i . The total contribution of SLP variations to surface net
longwave change is -0.028 W/m^2 . This is negligible compared to the 1.32 W/m^2 increase in net surface
flux during the same time period.

Figure 3. Wintertime (DJFM) local sea level pressure anomalies over time. Linear regression has a p-
value of 0.86 and an r-value of 0.02 (no trend).

Figure 4. Change in local sea level pressure anomalies from first vs. last 10 winters (DJFM) of
observations (left panel). Boxplot of net surface longwave radiation anomalies for each SLP bin range.

• Is there enough detail provided in the methods for the work to be reproduced?

Line 114: A few notes about the observations from NSA being used would be helpful. Obviously these
publications are very limited in length, but a sentence here describing the specific observation/product
would help with reader comprehension. There is a lot of detail under section 6, but given that ARM data
is not frequently used for these kinds of studies it would be helpful to have a few sentences in the main
text characterizing the observations being used and then the reader can dig in in the methods.

**Response:** We agree. We rewrote lines 177-180 to include several sentences about the observations
and methods within the main text.

**Reviewer #3 (Remarks to the Author)**

I co-reviewed this manuscript with one of the reviewers who provided the listed reports. This is part of
the Nature Communications initiative to facilitate training in peer review and to provide appropriate
recognition for Early Career Researchers who co-review manuscripts.

**Response:** We thank Reviewer #3 for their contributions to this review process.

Reviewer #4 (Remarks to the Author)

This paper presents a comprehensive analysis of the winter surface radiation budget and how this has
evolved as a consequence of Arctic climate warming, specifically how clouds have changed to affect this.
To my mind represent the first time this has been analyzed using direct measurements, not inferred
from satellite retrievals (which are observations but indirect) and not from modeling – which is iffy in
the Arctic where models struggle to represent even the current climate processes let alone how they
change. The results are very interesting and the arguments appears solid; to me it was a surprise that
the largest effect came from more opaque ice-only clouds. I think the current wisdom has more focused
on increasing fraction of liquid in mixed-phase clouds and hence these have been a focus for cloud
physics research; this study indicate that a larger focus should be on ice clouds. These distinctions are
important to get right in models, especially since many models tends to overestimate either liquid or ice.
The argument is that it is the ice clouds that holds potential to become more optically thick, while clouds
containing liquid would to a larger extent already behave like black bodies while the changes in the
occurrence of liquid-bearing clouds does not compensate for this, seems logical and in my mind is the
take-home message of the paper, besides the fact that clouds do matter for the Arctic amplified
warming.

**Response:** We thank Reviewer #4 for their thorough and largely positive review. We agree with the
reviewer’s assessment of our take-home message and its relation to the literature.

However, the paper is not particularly well organized nor well written, making it hard to follow the
arguments and some details are missed all together, like the fact that some of the attribution rests on
radiative transfer modelling and not on the observations; I didn’t get that until I read the methods
section. It is not that I object; more that I would have wanted to know. I think the manuscript needs a
real work-over before it is published and therefore I recommend major revision.

**Response:** We thank the reviewer for the opportunity to improve the communication of our results.
When specific suggestions have been provided, we have made changes in response to when reviewer #4
found our manuscript difficult to follow or unclear. That said, we did not substantially restructure our
paper. We note that reviewer #2 called our initial submission “very nicely written and easy to follow”
and reviewer #1 called it “a well-written article, with solid scientific logic”.

In response to this comment, we have mentioned the use of radiative transfer modelling in the main
text of the article in lines 177-180. That said, the radiative transfer modeling is done solely using
observations. This is typical of many attribution studies (e.g., Kay & L’Ecuyer (2013) Table 4, many other
studies, such as Kramer et al., 2019; Raghuraman et al., 2023, etc.).

Major comments:
I think the author needs to think more about the narrative. A paper is not just the results and the
methods; it needs to tell a story and it should be a story that is easy to read and grasp as far as is
possible. The main problem I had with this paper is not understanding how the different classifications
where accomplished. I did get that when I read the methods section but by the the damage was already
done.

**Response:** We agree with the reviewer that we could improve to story that we are telling. We have
rewritten the introduction to flow more smoothly. We introduced the cloud-phase classification in the

main text and added a reference forward to the appropriate section of the methodology for further
detail (line 214).

The order of presentation also matters; start with a hypothesis (why), then a brief description of what
the classes are (how) and then the results. In this paper there are many places where a section is started
with the conclusions followed by the results and then discussion but there is no background; no
hypothesis and no “why”. This paper also has a lot of unnecessary repetitions where something is
stated, then shown how, and then the same thing is restated again at the end of the same argument.
There is very little background; for example what the different feedback mechanisms are or what we
would expect to see; there are no hypotheses and to understand how the conclusions were drawn one
have to wait until the methods section.

**Response:** We agree that we could provide additional background and hypothesis in our analysis. To
that effect, we added background and hypothesis for each section of the paper where it was missing.
Net flux distribution background/hypotheses were added on lines 103-109 and 141-143. Non-cloud and
cloud driver background/hypotheses were added on lines 215-223.

The record lengths are skipped over rather carelessly on page two; over two decades of radiation
measurements and a decade of cloud observations. But the cloud observations are crucial for the
conclusions in the paper and those are only – what – 10 years? So exactly how long records are used for
what parts of the results? This needs to be up front and center; I have no problems with the trend
analyses of the radiation data, but the conclusions rest on 10 years of cloud data; barely enough to
define the state of things let alone a trend.

**Response:** We agree that we were imprecise about our cloud record duration. We have revised the text
to state it more clearly (lines 80-81). The cloud record duration is 11 winter seasons (DJFM) spanning 13
574 years: 2004-2006, 2008-2011, and 2014-2019.

We disagree with the reviewer 13 years is “barely enough to define the state of things”. We also
emphasize that we do not examine any secular trends in the cloud observations. We examine the
response to monthly temperature anomalies, which is less sensitive to the record duration than secular
trend analysis. For example, repeating our analysis without years 2004-2006 of cloud observations has
no effect on our conclusions.

There is also a lot of discussion of significant and non-significant trends or differences, but I could not
find anything about how the significance was tested.

**Response:** We agree that more information about significance testing is useful. We have added it to the
figure captions and methods so that it doesn’t distract from the flow of the text. Specifically, we have
added definitions of our treatment of uncertainty in section 6.4.2, lines 574-587. We also added an
indication in the caption that bars in Figure 3 report the 95% confidence interval.

**Minor comments:**

Line 38: Here increased opacity of ice-only and mixed phase clouds contribute equally, but that is not
what it looks like in the text; there it seems that ice-only clouds have a larger contribution.

**Response:** We thank the reviewer for pointing out this ambiguity. We have revised both the abstract
and lines 260-276. We now highlight that we examine increasing cloud opacity with and without a

change in cloud phase. Considering increasing cloud opacity overall, ice-only and liquid-containing
clouds drive equal increases in surface downward longwave radiation. Without considering a change in
cloud phase, ice-only clouds drive a larger increase because of the tendency of these clouds to originally
be less opaque than liquid containing clouds.

L 40: What do you mean by “sets the sign of”?

**Response:** We thank the reviewer for pointing out this unclear wording. We have changed this sentence
(lines 40-42) to now read: "The direct observational constraint from this work suggests that increasing
cloud opacity drives increasing net surface radiation on the North Slope of Alaska."

L 50: If “trapping” of heat below the clouds was the process here, how could that not show up at TOA
and be measured by satellite? If heat is trapped, there has to be less escaping to space?

**Response:** We thank the reviewer for highlighting the word “trapping” as apparently contradicting the
idea that both surface and TOA radiation must be considered. To address this point, we have changed
the word “traps” to “redistributes”.

Nonetheless, Arctic clouds do redistribute energy downwards in the atmosphere and warm the surface
without necessarily reducing TOA radiation (Li et al., 2023; Sedlar et al., 2012). The relative
independence of TOA and surface radiation can also be seen from ARM NSA observations (Figure 5).
TOA outgoing longwave radiation can remain constant for a wide (~70 W/m²) range of surface
downwelling longwave fluxes, which would have a pronounced impact on surface temperatures without
a reduction in TOA radiation.

Figure 5. Wintertime ARM NSA distributions of surface downwelling longwave binned by top-of-
atmosphere outgoing longwave radiation (OLR). Bins are spaced at 10 W/m², and lines indicate 5th and
95th percentiles. Text indicates difference between 95th and 5th percentiles for each OLR bin.

L 55-53: Correct me if I'm wrong, but the warming of the Arctic must be a consequence of the mismatch
between heat advected into the region from south and changes in the TOA. That defines climate; not
what happens at the surface. So why should factors contributing to climate change not use TOA? If
clouds in the Arctic were to warm the surface more, then this begs the question from where comes that
energy? The problem with passive sensor satellites is that they needs retrievals that are notoriously
unreliable especially in the Arctic; not that they measure the wrong thing. Moreover, many of the

problems with active sensor satellites are mostly due to design choices – they miss most of the low
clouds which happens to be dominant in the Arctic, and they just barely reach into the souther parts of
the Arctic.

**Response:** We agree that the reviewer’s viewpoint constrains the total climate feedback, but we do not
think it can constrain individual feedbacks. What about a feedback process that is positive at the surface
but negative at the TOA? Would we describe this as a feedback that amplifies warming or dampens it?
Goosse et al. (2018) describes such a scenario:

"Since TOA fluxes determine the total energy budget of the Earth’s climate system, they are a
natural reference point for computing climate feedbacks at a global scale. They are also closely
connected to surface temperature change in the Tropics, where deep convection leads to a
vertically well-mixed atmosphere. In the Arctic, where deep vertical mixing is suppressed by
strong static stability in the troposphere, computing feedback parameters based on surface
fluxes can lead to important additional insights. For example, a change in clouds that raises
atmospheric emissivity in the Arctic inversion layer can lead to increases in both upwelling and
downwelling longwave radiation, and thus lead to energy loss and a negative cloud feedback at
TOA but energy gain and a positive cloud feedback at the surface."

Since we are interested in constraining an individual feedback process (i.e. the cloud response to
warming’s feedback effect on surface temperature), we consider surface radiation since it is more
directly related to local warming.

L 74-77: The fact that others have used short records is a poor excuse; did you look into the effects of
the short record? Also what record length is used for the cloud data?

**Response:** We did not change the manuscript in response to this comment. We disagree that using a
record length consistent with a decade of prior literature is a “poor excuse”. This prior literature (see
citations in L74-77 of original manuscript) has provided valuable insights that have improved
understanding of cloud feedbacks.

Nonetheless, we did investigate the effects of the short record. Repeating our analysis with an artificially
shortened record (2008 onwards) had no effect on our conclusions. In other words, our conclusions are
insensitive to a slightly shorter (and likely thus a slightly longer) record.

See lines 572-574 of this document for the record length of the cloud data. We emphasize that our
choice to focus on responses to temperature anomalies rather than secular trends makes our analysis
less sensitive to the record length.

Section starting at line 134: I would like to see an expanded discussion here, especially of how changes
in cloud phase, cloud fractions and temperature would affect the local minimum in the distribution of
net radiation.

**Response:** We thank the reviewer for highlighting a point where our presentation could be more
detailed. We have added an additional sentence in this paragraph qualitatively summarizing the cloud
phase, altitude, and cover associated with each mode. Now lines 136-141 read "The radiatively clear
state is associated with cold air and thin ice clouds or clear skies, while the opaquely cloud state is
associated with relatively warmer air and overcast mixed-phase clouds [15, 41–45]".

Is the same value used to distinguish clear and cloudy even when temperature and clouds change? On L
176-177 you state they do not change. I wonder why and is that evident?

**Response:** We believe that these concerns are already sufficiently addressed in the manuscript, so we
have not made any changes in response to this comment. Radiation thresholds are not used for any part
of our cloud observational product or driver analysis. As stated in lines 433-434, cloud observations rely
on radar, lidar, ceilometer, microwave radiometer, and radiosonde. This list does not include infrared
radiometer.

Additionally, in the original manuscript, we stated that “warming and increasing greenhouse gases alone
(Methods, Sec. 6.3) have no effect on opaque mode frequency”. As outlined in methods section 6.3, we
recalculated the radiation with a warmer atmosphere and found no change in the opaque mode (Fig. 1c,
dashed line).

Line 213: What about sensible heat?

**Response:** We thank the reviewer for pointing out additional non-radiative terms of the surface energy
budget that we should consider. We revised this portion of the manuscript (lines 167-174) to make a
simpler argument: since the Planck response explains the observed upwelling longwave flux increase
(Figure 2 on line 142 of this document), the sum of changes to all other non-radiative fluxes (sensible
heat, latent heat, subsurface heat flux) must be small.

We did investigate sensible heat changes at ARM NSA to provide another check on this line of argument.
This new investigation found no evidence that increasing sensible heat suppresses the upwelling
longwave response to warming (Figure 6). Sensible heat does not have a significant response to monthly
temperature anomalies, and neither does the closely related surface-air temperature contrast. This adds
a level of confidence in our assessment that changes to non-radiative surface fluxes are likely small in
comparison to the radiative flux response to warming.

Figure 6. Sensible heat change over time and with warming from two perspectives. First row: eddy-covariance derived sensible heat (ARM 2011) vs. time and SAT anomaly. Second row: difference between surface skin and near-surface air temperatures vs. time and SAT anomaly.

L 217: Just a reference forward in the text doesn't hack it; you need some description of it here for the results to mean anything. No one is going jump forward and read the methods section before reading the rest of the text first.

Response: We thank the reviewer for the suggestion on how to improve the readability of the manuscript. We rewrote lines 179-214 to include several sentences about the observations and methods within the main text.

L 220: What do you mean by "only explains 76%"; that seems like a lot to me?

Response: We thank the reviewer for highlighting this confusing word choice. We removed "only" from the sentence (line 226).

L 228-229: How "suppress"? What do you man by this?

Response: We thank the reviewer for highlighting this point of confusion. We have removed the word "suppress" from the sentence.

L 254-258: This is mostly a repetition to things already stated.

**Response:** We rewrote this section of the article in response to an earlier comment by the reviewer
(lines 551-558 of this document). In our rewrite we avoided repetitive statements.

L 260-263: The second of these sentences seems like a repletion of the first.

**Response:** We rewrote this section of the article in response to an earlier comment by the reviewer
(lines 551-558). In our rewrite we avoided repetitive statements.

Section 4: Is surface pressure really a good metric for circulation variability. Think about a cyclone
passing northward to your east or to your west. The pressure could be identical and yet one would
represent advection of dry and cold Arctic air while the other would bring moist warm mid-latitude air.
This section is not very convincing.

**Response:** We agree that local SLP does not have a necessary connection to warm vs. cold air advection.
In response to the reviewer concern, we further investigated the suitability of local SLP as a circulation
confounder variable. We found evidence that SLP is suitable as a confounding variable for four reasons.
First, we note that SLP has clear relationships with temperature, net surface radiative flux, and cloud
water path at ARM NSA (Figure 7 a, c, e). Second, local SLP has been used as a confounding variable in
previous studies (e.g., Morrison et al. 2012 review paper on Arctic clouds). Third, local SLP is a direct
observable while an estimate of advective temperature tendency relies on models/reanalysis data.
Finally, we did an addition analysis to assess a confounding variable with clearer physical meaning for
advection than local SLP (see lines 756-767, Figure 8 below for details). We found using this new variable
gave the same result as for local SLP. Given these reasons, we retained local SLP did not revise the
original analysis in response to this comment.

This paragraph provides more information on the circulation confounder analysis we did for a variable
more related to advection than local SLP. In doing this analysis, we found the same result for this new
variable and for SLP (Figure 8). Specifically, the new variable we used was daily standard deviation of
near-surface (1000-900 hPa) advective temperature tendency from the ERA5 reanalysis. Variance in
advective temperature tendency is easier to interpret as synoptic disturbances (e.g., frontal passages)
than local SLP. But like local SLP, advective temperature tendency variance also has significant ($p < 0.01$)
relationships with temperature, radiation, and cloud water path (Figure 7 b,d,f). Stratifying by this
variable shows that net flux and cloud water path still increase independent of circulation variability. In
other words, our circulation confounder results are similar for local SLP and for this new reanalysis-
based advection variable. This similarity provides strong support for us to retain the use of local SLP as
originally written. Nevertheless, we included this additional analysis in the supplement and referenced it
in the main text.

Figure 7. Daily relationships between confounder variables and quantities used in manuscript Figure 4. Near-surface air temperature anomaly (a,b), net surface longwave flux (c,d), and total water path (e,f) vs. local sea level pressure anomaly (a,c,e) and standard deviation of advective temperature tendency from reanalysis.

Figure 8. Like manuscript Figure 4 but using daily standard deviation of advective temperature tendency
instead of local surface pressure. (a-b) Net surface flux and (c-d) cloud water path response to warming
with (a,c) and without (b,d) controlling for circulation variability. (e) Frequency of each local advective
temperature tendency bin. Bars show 95% confidence interval for regression slopes. (c) Total and
individual ice and liquid contributions to cloud water path.

L 393-394: Are soundings also included here?

**Response:** Yes, the Shupe-Turner microphysics retrieval includes radiosonde observations. We thank the
reviewer for this point and have added it to the list of instruments.

References

- Atmospheric Radiation Measurement (ARM) user facility. 1997. European Centre for Medium Range
Weather Forecasts Diagnostic Analyses (ECMWFTE), 1997-09-01 to 2024-10-01, North Slope
Alaska (NSA) External Data (satellites and others) (X1). ARM Data Center. Data set accessed
2025-06-22.
- Atmospheric Radiation Measurement (ARM) user facility. 1997. European Centre for Medium Range
Weather Forecasts Diagnostic Analyses (ECMWFVAR), 1997-09-01 to 2024-10-01, North Slope
Alaska (NSA) External Data (satellites and others) (X1). ARM Data Center. Data set accessed
2025-06-22.
- Atmospheric Radiation Measurement (ARM) user facility. 2011. Quality Controlled Eddy Correlation Flux
Measurement (30QCECOR), 2011-11-07 to 2024-09-30, North Slope Alaska (NSA) Barrow,
Alaska, extended facility colocated at main site (E10). Compiled by K. Gaustad. ARM Data
Center. Data set accessed 2025-06-23 at <http://dx.doi.org/10.5439/1097546>.
- Barrientos-Velasco, C., Cox, C. J., Deneke, H., Dodson, J. B., Hünerbein, A., Shupe, M. D., Taylor, P. C., &
Macke, A. (2024). *Estimation of the radiation budget during MOSAiC based on ground-based and*
*satellite remote sensing observations*. <https://doi.org/10.5194/egusphere-2024-2193>
- Barrientos-Velasco, C., Deneke, H., Hünerbein, A., Griesche, H. J., Seifert, P., & Macke, A. (2022).
Radiative closure and cloud effects on the radiation budget based on satellite and shipborne
observations during the Arctic summer research cruise, PS106. *Atmospheric Chemistry and*
*Physics*, 22(14), 9313–9348. <https://doi.org/10.5194/acp-22-9313-2022>
- Ebell, K., Nomokonova, T., Maturilli, M., & Ritter, C. (2020). Radiative Effect of Clouds at Ny-Ålesund,
Svalbard, as Inferred from Ground-Based Remote Sensing Observations. *Journal of Applied*
*Meteorology and Climatology*, 59(1), 3–22. <https://doi.org/10.1175/JAMC-D-19-0080.1>
- Goosse, H., Kay, J. E., Armour, K. C., Bodas-Salcedo, A., Chepfer, H., Docquier, D., Jonko, A., Kushner, P.
844 J., Lecomte, O., Massonnet, F., Park, H.-S., Pithan, F., Svensson, G., & Vancoppenolle, M. (2018).
Quantifying climate feedbacks in polar regions. *Nature Communications*, 9(1), 1919.
<https://doi.org/10.1038/s41467-018-04173-0>
- Graham, R. M., Rinke, A., Cohen, L., Hudson, S. R., Walden, V. P., Granskog, M. A., Dorn, W., Kayser, M.,
& Maturilli, M. (2017). A comparison of the two Arctic atmospheric winter states observed
during N-ICE2015 and SHEBA. *Journal of Geophysical Research: Atmospheres*, 122(11), 5716–
5737. <https://doi.org/10.1002/2016JD025475>
- Griesche, H. J., Barrientos-Velasco, C., Deneke, H., Hünerbein, A., Seifert, P., & Macke, A. (2024). Low-
level Arctic clouds: A blind zone in our knowledge of the radiation budget. *Atmospheric*
*Chemistry and Physics*, 24(1), 597–612. <https://doi.org/10.5194/acp-24-597-2024>
- Kay, J. E., & L'Ecuyer, T. (2013). Observational constraints on Arctic Ocean clouds and radiative fluxes
during the early 21st century. *Journal of Geophysical Research: Atmospheres*, 118(13), 7219–
7236. <https://doi.org/10.1002/jgrd.50489>

- Kramer, R. J., Matus, A. V., Soden, B. J., & L'Ecuyer, T. S. (2019). Observation-Based Radiative Kernels
From CloudSat/CALIPSO. *Journal of Geophysical Research: Atmospheres*, *124*(10), 5431–5444.
<https://doi.org/10.1029/2018JD029021>
- Lelli, L., Vountas, M., Khosravi, N., & Burrows, J. P. (2023). Satellite remote sensing of regional and
seasonal Arctic cooling showing a multi-decadal trend towards brighter and more liquid clouds.
*Atmospheric Chemistry and Physics*, *23*(4), 2579–2611. [https://doi.org/10.5194/acp-23-2579-](https://doi.org/10.5194/acp-23-2579-2023)
[2023](https://doi.org/10.5194/acp-23-2579-2023)
- Li, X., Mace, G. G., Strong, C., & Krueger, S. K. (2023). Wintertime Cooling of the Arctic TOA by Low-Level
Clouds. *Geophysical Research Letters*, *50*(17), e2023GL104869.
<https://doi.org/10.1029/2023GL104869>
- Maturilli, M., Herber, A., & König-Langlo, G. (2015). Surface radiation climatology for Ny-Ålesund,
Svalbard (78.9° N), basic observations for trend detection. *Theoretical and Applied Climatology*,
*120*(1–2), 331–339. <https://doi.org/10.1007/s00704-014-1173-4>
- Morrison, H., Boer, G. de, Feingold, G., Harrington, J., Shupe, M. D., & Sulia, K. (2012). Resilience of
persistent Arctic mixed-phase clouds. *Nature Geoscience*, *5*(1).
<https://doi.org/10.1038/ngeo1332>
- Mülmenstädt, J., Lubin, D., Russell, L. M., & Vogelmann, A. M. (2012). Cloud Properties over the North
Slope of Alaska: Identifying the Prevailing Meteorological Regimes. *Journal of Climate*, *25*(23),
8238–8258. <https://doi.org/10.1175/JCLI-D-11-00636.1>
- Nakoudi, K., Ritter, C., & Stachlewska, I. S. (2021). Properties of Cirrus Clouds over the European Arctic
(Ny-Ålesund, Svalbard). *Remote Sensing*, *13*(22), 4555. <https://doi.org/10.3390/rs13224555>
- Nomokonova, T., Ebell, K., Löhnert, U., Maturilli, M., & Ritter, C. (2020). The influence of water vapor
anomalies on clouds and their radiative effect at Ny-Ålesund. *Atmospheric Chemistry and*
*Physics*, *20*(8), 5157–5173. <https://doi.org/10.5194/acp-20-5157-2020>
- Persson, P. O. G. (2002). Measurements near the Atmospheric Surface Flux Group tower at SHEBA: Near-
surface conditions and surface energy budget. *Journal of Geophysical Research*, *107*(C10), 8045.
<https://doi.org/10.1029/2000JC000705>
- Raddatz, R. L., Papakyriakou, T. N., Else, B. G., Asplin, M. G., Candlish, L. M., Galley, R. J., & Barber, D. G.
(2015). Downwelling longwave radiation and atmospheric winter states in the western maritime
Arctic: ATMOSPHERIC WINTER STATES IN THE WESTERN MARITIME ARCTIC. *International*
*Journal of Climatology*, *35*(9), 2339–2351. <https://doi.org/10.1002/joc.4149>
- Raghuraman, S. P., Paynter, D., Menzel, R., & Ramaswamy, V. (2023). Forcing, Cloud Feedbacks, Cloud
Masking, and Internal Variability in the Cloud Radiative Effect Satellite Record. *Journal of*
*Climate*, *36*(12), 4151–4167. <https://doi.org/10.1175/JCLI-D-22-0555.1>
- Sedlar, J., Shupe, M. D., & Tjernström, M. (2012). On the Relationship between Thermodynamic
Structure and Cloud Top, and Its Climate Significance in the Arctic. *Journal of Climate*, *25*(7).
<https://doi.org/10.1175/JCLI-D-11-00186.1>

- Shupe, M. D., Turner, D. D., Zwink, A., Thieman, M. M., Mlawer, E. J., & Shippert, T. (2015). Deriving
Arctic Cloud Microphysics at Barrow, Alaska: Algorithms, Results, and Radiative Closure. *Journal*
*of Applied Meteorology and Climatology*, 54(7), 1675–1689. [https://doi.org/10.1175/JAMC-D-](https://doi.org/10.1175/JAMC-D-15-0054.1)
15-0054.1
- Solomon, A., Shupe, M. D., Svensson, G., Barton, N. P., Batrak, Y., Bazile, E., Day, J. J., Doyle, J. D., Frank,
H. P., Keeley, S., Remes, T., & Tolstykh, M. (2023). The winter central Arctic surface energy
budget: A model evaluation using observations from the MOSAiC campaign. *Elem Sci Anth*,
11(1), 00104. <https://doi.org/10.1525/elementa.2022.00104>
- Stramler, K., Del Genio, A. D., & Rossow, W. B. (2011). Synoptically Driven Arctic Winter States. *Journal*
*of Climate*, 24(6), 1747–1762. <https://doi.org/10.1175/2010JCLI3817.1>
- Wendisch, M., Brückner, M., Crewell, S., Ehrlich, A., Notholt, J., Lüpkes, C., Macke, A., Burrows, J. P.,
Rinke, A., Quaas, J., Maturilli, M., Schemann, V., Shupe, M. D., Akansu, E. F., Barrientos-Velasco,
C., Bärfuss, K., Blechschmidt, A.-M., Block, K., Bougoudis, I., ... Zeppenfeld, S. (2023).
Atmospheric and Surface Processes, and Feedback Mechanisms Determining Arctic
Amplification: A Review of First Results and Prospects of the (AC)3 Project. *Bulletin of the*
*American Meteorological Society*, 104(1), E208–E242. [https://doi.org/10.1175/BAMS-D-21-](https://doi.org/10.1175/BAMS-D-21-0218.1)
0218.1
- Yamanouchi, T. (2019). Arctic warming by cloud radiation enhanced by moist air intrusion observed at
Ny-Ålesund, Svalbard. *Polar Science*, 21, 110–116. <https://doi.org/10.1016/j.polar.2018.10.009>

Reviewer #1 (Remarks to the Author)

The authors have done a considerable job in addressing my remarks. As far as I am concerned, the paper can be published as it is.

Still, at authors' discretion, a closing speculative sentence in the discussion section on future steps is welcome.

To be precise: according to the authors, what are the current limitations of their study and what can be done to extend the presented results both in terms of time (other seasons) and in terms of methodology and data? In fact, a concluding message to the community on what the authors believe is necessary to further explore the topic.

Response: We thank reviewer #1 for their time considering our responses and the revised manuscript. In response to this comment, we have added a closing statement on lines 421-424: "Future work might develop along two lines of inquiry – what cloud processes drive increased ice production and increasing cloud opacity generally? Does a thin-to-opaque cloud transition occur in other seasons and regions of the Arctic, or is it unique to NSA winter?"

**Reviewer #2 (Remarks to the Author)**

I thank the authors for their time thoroughly responding to my questions. I have no further suggestions
or questions at this time.

**Response:** We thank reviewer #2 for their time considering our responses and the revised manuscript.

Reviewer #3 (Remarks to the Author)

Response: We thank Reviewer #3 for their contributions to this review process.

Reviewer #4 (Remarks to the Author)

This is a revised version of a study that – with the exception of using a radiative transfer code – uses
ground-based observations to draw conclusions on cloud feedbacks on surface energy fluxes in the
Arctic. If my previous review came across as critical, let me start here by saying that I find this work
ingenious and novel and with some important, to me somewhat surprising, conclusions; it should be
published. At this time I also want to congratulate the authors on the revision, that has made the
manuscript much more enjoyable to read. The reservations I had for it lacking a narrative are alleviated.
Also on the quite a few detailed comments, the authors have made a through job. Although we are not
in complete agreement on all items, I will write this down as a matter of opinion.

**Response:** We thank reviewer #4 for their time considering our responses and the revised manuscript.

Hence, it is my recommendation at this time is that it be published after a minor revision. I have,
however, a few remaining comments that I would wish the authors to consider:

I find the use of the term “phase” somewhat confusing and I would recommend that the authors use
“cloud type” rather than “cloud phase” whenever they discuss the three main types of clouds: “ice-
only”, mixed-phase” and “liquid-only”. They should use the term “phase” only when discussing the
effects of changes in phase. This is anyway how I interpret the results presented in Figure 2, and the
discussion relating to it; that the 3rd column from the right in Figure 2e refers to effects of changes in
cloud phase, i.e. ice clouds more often becoming mixed-phase or liquid only with the warming. As the
discussion turns to Figure 3, the word “phase” now seem to change meaning to “type”; at least that is
how I understand the caption to Figure 3, where “+Clouds” now mean all the 4 effects for the different
“types” (= “phases”?). After all, cloud water only has two phases; liquid or ice. And vapor of course, but
then its not a cloud anymore’...

**Response:** We thank the reviewer for highlighting this ambiguity in our language. We have clarified this
point at several places in the text. We added the following to the caption of figure 3: “All-sky net flux
change (a) includes changes in both cloud phase type frequencies and properties (Eq. 4). Net flux change
for an individual cloud phase type (b-e) only includes changing properties.” We have also changed our
language to “cloud phase type” when referring to our decomposition of drivers according to cloud phase
and “cloud phase change” when referring to the effect of changing frequency of types on all-sky
radiation.

On Lines 348-350:

Still, the three lowest-pressure bins has a significantly larger effects than the three highest-pressure
bins. Also, I would not use the word “confound” here; you find that the temperature anomalies better
explain the cloud effects than the pressure anomalies, but the temperature anomalies themselves
comes from somewhere, possibly from changes in large scale advection. So maybe the local effects are
temperature driven but the temperature changes are – in part – “weather driven”.

**Response:** We disagree with the reviewer’s assessment of our analysis. The three lowest-pressure bins
do not have significantly larger correlations than the three highest-pressure bins at the 95% confidence
level. As can be seen from Figure 3, the confidence intervals overlap between all the comparisons which

can be made between high- and low-pressure bins. While it is possible that significant differences may
be detected from additional analysis, our analysis makes no such differences clearly apparent.

Furthermore, we believe that our use of the term “confounding” is appropriate. We seek to infer the
casual effect of long-term warming on cloud opacity. Thus, we seek to rule out the possibility that a third
extraneous variable (short-term circulation variability) causes temperature and cloud opacity to increase
independent from long-term change. In other words, short-term circulation variability could confound
our attempt to infer a long-term causal relationship. A recent example of this kind of confounding in
observational estimates of climate processes can be seen in Mülmenstädt et al. (2024).

To clarify this point, we rewrote lines 341-343 to read “As such, short-term variations in the frequency
of synoptic disturbances might confound the long-term cloud and radiation response to warming
estimated above.” With this change, we hope to make clear that we are not ruling out all circulation
influence on climate change at ARM NSA. We simply want to rule out climatological variability as the
cause of the temperature-cloud-radiation relationship we investigate.

*Line 366-368: I still struggle with the conceptualization of climate and climate change here. Bottom line,*
*climate change in a region is due to a mismatch between net radiation at TOA and import of heat over*
*the boundaries. Now you say that temperature increase may be additionally enhanced by local changes,*
*in clouds, a radiatively driven additional warming (“accumulation of surface energy”) near the surface*
*from changing clouds that is not detectable from space? So where does that energy come from, if not*
*from advection?*

**Response:** We thank the reviewer for highlighting ambiguity in our language. To state our position
clearly: we agree that Arctic climate is a balance between radiative cooling at TOA and advective
warming from lower latitudes. We do not claim that advection of heat into the Arctic is constant with
warming. In our confounding analysis, we simply examine whether short-term fluctuations in local
SLP/temperature advection explain our results.

We have revised the relevant portion of the discussion (lines 402-406) to be more precise – “In other
words, increasing surface temperatures do not increase surface radiative cooling because of changing
cloud properties. This fact provides direct observational evidence that clouds contribute to surface-
amplified warming during winter at NSA.”

*On Lines 463-473: While I’m not necessarily questioning the use of RRTM-LW as such, and I’m not*
*worried by the comparison statistics between modelled and observed radiation either; its well within*
*the measurement uncertainty. However, I’m still curious; isn’t there a consistent (except the very last*
*years) negative bias, or maybe even a positive trend in bias? And what happened with the missing years*
*in Figure 5a? What is the reason for the distribution in Figure 5b being the “fattest” for mid-range*
*values, around 180 Wm⁻²? Are those cases with a broken cloud field? And how do you deal with that in*
*the model; care to comment?*

**Response:** We agree with the reviewer that our degree of radiative closure is sufficient for our analysis.

Nevertheless, we answer the queries about our closure analysis. We agree that our calculations tend to
have a minor negative bias rather than be evenly distributed around zero. This is likely because the
entire atmospheric state cannot always be observed and some interpolation between observations is
necessary. While a positive trend in bias is possible, it may also be due to the small (n = 13) number of

samples. A future study should determine whether this occurs in winter alone or is present in all
seasons. The reviewer is correct in speculating that the widest distribution in Figure 5b occurs for cases
with a broken cloud field. RRTM-LW is a one-dimensional radiative transfer model and thus cannot
incorporate the three-dimensional cloud field which affects observed surface radiative fluxes. This is
accounted for by exchanging an average in space for an average in time – broken cloud fields become
less problematic when we are only examining perturbations about the (time-averaged) mean state in
our analysis.

**References**

Mülmenstädt, J., Gryspeerdt, E., Dipu, S., Quaas, J., Ackerman, A. S., Fridlind, A. M., Tornow, F., Bauer, S.
E., Gettelman, A., Ming, Y., Zheng, Y., Ma, P.-L., Wang, H., Zhang, K., Christensen, M. W., Varble,
286 A. C., Leung, L. R., Liu, X., Neubauer, D., Partridge, D. G., Stier, P., and Takemura, T.: General
circulation models simulate negative liquid water path–droplet number correlations, but
anthropogenic aerosols still increase simulated liquid water path, *Atmos. Chem. Phys.*, 24,
7331–7345, <https://doi.org/10.5194/acp-24-7331-2024>, 2024.